# Tocotrienol-Rich Vitamin E (Tocovid) Improved Nerve Conduction Velocity in Type 2 Diabetes Mellitus Patients in a Phase II Double-Blind, Randomized Controlled Clinical Trial

**DOI:** 10.3390/nu13113770

**Published:** 2021-10-25

**Authors:** Pei Fen Chuar, Yeek Tat Ng, Sonia Chew Wen Phang, Yan Yi Koay, J-Ian Ho, Loon Shin Ho, Nevein Philip Botross Henien, Badariah Ahmad, Khalid Abdul Kadir

**Affiliations:** Jeffrey Cheah School of Medicine and Health Sciences, Monash University Malaysia, Jalan Lagoon Selatan, Bandar Sunway, Subang Jaya 47500, Selangor, Malaysia; danny2tng@gmail.com (Y.T.N.); sonia.phang1@monash.edu (S.C.W.P.); yanyikoay@gmail.com (Y.Y.K.); ian.ho1996@gmail.com (J.-I.H.); ho.loon.shin@monash.edu (L.S.H.); nevein.botross@monash.edu (N.P.B.H.)

**Keywords:** type 2 diabetes mellitus (T2DM), diabetic peripheral neuropathy, tocotrienol, vitamin E, transforming growth factor beta-1 (TGFβ-1), vascular endothelial growth factor A (VEGF-A)

## Abstract

Diabetic peripheral neuropathy (DPN) is the most common microvascular complication of diabetes that affects approximately half of the diabetic population. Up to 53% of DPN patients experience neuropathic pain, which leads to a reduction in the quality of life and work productivity. Tocotrienols have been shown to possess antioxidant, anti-inflammatory, and neuroprotective properties in preclinical and clinical studies. This study aimed to investigate the effects of tocotrienol-rich vitamin E (Tocovid Suprabio^TM^) on nerve conduction parameters and serum biomarkers among patients with type 2 diabetes mellitus (T2DM). A total of 88 patients were randomized to receive 200 mg of Tocovid twice daily, or a matching placebo for 12 months. Fasting blood samples were collected for measurements of HbA1c, renal profile, lipid profile, and biomarkers. A nerve conduction study (NCS) was performed on all patients at baseline and subsequently at 2, 6, 12 months. Patients were reassessed after 6 months of washout. After 12 months of supplementation, patients in the Tocovid group exhibited highly significant improvements in conduction velocity (CV) of both median and sural sensory nerves as compared to those in the placebo group. The between-intervention-group differences (treatment effects) in CV were 1.60 m/s (95% CI: 0.70, 2.40) for the median nerve and 2.10 m/s (95% CI: 1.50, 2.90) for the sural nerve. A significant difference in peak velocity (PV) was also observed in the sural nerve (2.10 m/s; 95% CI: 1.00, 3.20) after 12 months. Significant improvements in CV were only observed up to 6 months in the tibial motor nerve, 1.30 m/s (95% CI: 0.60, 2.20). There were no significant changes in serum biomarkers, transforming growth factor beta-1 (TGFβ-1), or vascular endothelial growth factor A (VEGF-A). After 6 months of washout, there were no significant differences from baseline between groups in nerve conduction parameters of all three nerves. Tocovid at 400 mg/day significantly improve tibial motor nerve CV up to 6 months, but median and sural sensory nerve CV in up to 12 months of supplementation. All improvements diminished after 6 months of washout.

## 1. Introduction

Diabetic peripheral neuropathy (DPN) is one of the most common microvascular complications of diabetes that affects roughly 50% of the type 2 diabetic population [1]. It is the most common cause of foot ulcers [2] and is involved in 50–75% of non-traumatic amputations [3]. Up to 25% of patients with DPN will eventually develop foot ulcers during their lifetime [3].

Diabetic neuropathic pain is hard to identify and treat because its pathophysiological mechanism is not yet fully understood. Up to 53% of patients with DPN experience neuropathic pain, reducing their quality of life and work productivity [4]. Currently, there is no known cure for diabetic neuropathy. Treatments that aim to slow its progression and relieve pain remain as the main therapeutic objective without addressing the underlying pathological mechanisms of DPN. However, neuropathic pain generally responds poorly to traditional painkillers [5,6].

The early diagnosis of diabetic peripheral neuropathy is difficult because half of the patients do not display any symptoms [7]. However, alternations in nerve conduction may be present even in early, asymptomatic stages of diabetic peripheral neuropathy. In recent years, electrophysiological techniques such as nerve conduction studies have become more popular and are widely used to identify subclinical pathological changes for the early discovery of DPN due to its objective and sensitive nature [8]. Recently, researchers found that whole plantar nerve (WPN) conduction studies are extremely useful for early detection of the length-dependent DPN [9]. Patients with DPN usually have a prolonged latency, slower conduction velocity, or lower amplitude of action potential in nerve conduction studies, and in extreme cases, the waveform is completely absent [8].

Numerous beneficial effects of tocotrienol on diabetes were documented in a systematic review by Pang K.L. et al. [10]. In this review of preclinical and clinical studies, tocotrienol was shown to provide kidney, cardiovascular, and nerve protection, present anti-diabetic, anti-inflammatory, antioxidant, and anti-cataractogenic effects, as well as improve lipid profiles and skeletal muscle health [10]. Tocotrienols share a similar structure with tocopherols, except for the presence of three unsaturated bonds at the side chains which provide more efficient penetration into tissues that have saturated fatty layers, such as the brain and liver [11]. Experimental research examining the antioxidant, free radical scavenging effects of tocopherols, and tocotrienols revealed that tocotrienols appear superior due to their better distribution in fatty layers of the cell membrane [11]. Oral administration of tocotrienols mixture was reported to reverse neuropathic pain in diabetic rats by reducing oxidative–nitrosative stress and caspase-3 activity, as well as inhibiting the release of inflammatory cytokines [12].

The primary objective of this clinical trial was to investigate the effect of tocotrienol-rich vitamin E (Tocovid Suprabio^TM^—hereafter referred to as Tocovid in this paper) on DPN, as assessed by changes in nerve conduction velocity and action potential amplitude obtained from nerve conduction studies in patients with type 2 diabetes mellitus (T2DM). Secondly, this clinical trial aimed to explore the effect of tocotrienol-rich vitamin E on serum biomarkers, namely, transforming growth factor beta-1 (TGFβ-1) and vascular endothelial growth factor A (VEGF-A). Lastly, we also determined whether the effects of tocotrienol-rich vitamin E on nerve function would persist after 6 months of washout.

## 2. Materials and Methods

### 2.1. Study Design

This study was initiated in 2019 as a double-blind, placebo-controlled, multicentre trial in 88 T2DM participants with good glycaemic control (HbA1c between 6.0% and 9.0%) to investigate the effect of Tocovid on diabetic neuropathy. At eight weeks of treatment, the code was unblinded to one of the researchers so as to perform a preliminary analysis of the efficacy of the treatment and whether to continue the trial. It was found that eight weeks of Tocovid supplementation at 200 mg twice a day significantly improved nerve conduction velocities for median, sural, and tibial nerves, as well as elevated levels of serum NGF [13]. Hence, the trial was continued, blinded to the rest of the researchers and the patients. The unblinded researcher completed their degree and left, whilst the rest continued with the trial. Patients from the study were followed up for another 10 months to complete 12 months of treatment. After 6 months of treatment, the patients were reassessed at the CRCs to determine whether the beneficial effects had continued or persisted after treatment cessation.

The patients who had given consent and fulfilled the inclusion criteria were randomized and given 200 mg of Tocovid twice a day or a matching placebo for 12 months, followed by a 6-month washout period. At screening, nerve conduction studies were conducted on all patients and the relevant parameters were recorded as baseline values. Nerve conduction studies and other biochemical parameters and biological markers were repeated at 2 months, 6 months, and 12 months of the study period, and after 6 months of washout.

### 2.2. Participants

Study participants were recruited from an existing pool of diabetic patients who come for regular follow-ups at the Clinical Research Centres (CRCs) of Monash University in Bandar Sunway and Clinical School Johor Bahru, Tanglin Health Clinic, and Thomson Hospital in Kota Damansara, Malaysia. We reviewed the patients on follow up at these respective centres, based on the protocol inclusion and exclusion criteria, to determine potential suitability for the study before reviewing them for formal screening for enrolment. Potential participants were informed about the study and given the explanatory notes and consent forms to consider their possible participation. Those who were interested were invited to visit Monash University’s CRCs for screening.

#### 2.2.1. Inclusion Criteria

Patients with T2DM aged from 35 to 75 years old with stable glycaemic control (HbA1c of 6% to 9% with no more than 10% changes over the previous 2 months) were eligible for the study. Patients with hypertension had to have had stable blood pressure control for the past 2 months (less than 10% change and blood pressure (BP) of less than 150/90 mmHg).

#### 2.2.2. Exclusion Criteria

Patients were excluded if they had taken other water-soluble antioxidants such as vitamin C, glutathione, or polyphenols in the past 2 weeks, or fat-soluble antioxidants such as vitamin A, D, E, and K in the past month. Those who were heavy smokers or had stopped smoking for less than 1 month were not eligible for the study. Patients with any recent severe or acute illness such as active cancer, acute coronary syndrome, liver diseases, or any inflammatory disorders were excluded. Patients with stage 4 or below chronic kidney disease (CKD), with estimated glomerular filtration rates (eGFR) of less than 30 mL/min/1.73 m^2^, were excluded. Pregnant women and patients with unstable eye diseases were also excluded.

### 2.3. Randomization and Blinding

Patients were randomized 1:1 into a double-blinded treatment period for 12 months. Stratification was performed based on gender (male or female), HbA1c levels (median of 8%), and the duration of diabetes (median of 15 years) by an independent consultant with a Microsoft Excel spreadsheet. Those who were allocated to the interventional group received 200 mg of Tocovid twice daily after meals, whereas those in the control group received a matching placebo. Both investigational products were sponsored by ExcelVite Pty Ltd. and manufactured by Hovid Pharmaceutical Berhad. The investigational products were labelled as Drug B and Drug D, and their identity was kept confidential by ExcelVite. Both the Tocovid and placebo capsules were identical in size, shape, and colour; thus, their identity was not disclosed. This was a double-blinded study; hence, the randomization and allocation remained concealed from all the investigators and participants until the end of the study. The dosage used in this study was the maximum dose approved by the U.S. Food and Drug Administration (FDA).

### 2.4. Procedures

#### 2.4.1. Screening Visit

Informed consent was obtained from the participants before the commencement of screening. A thorough history-taking and physical examination were performed to ensure that participants met the inclusion/exclusion criteria. Patient’s blood pressure and anthropometric measurements such as weight, height, and waist circumference were taken. Urine samples were collected for full and microscopic examination (FEME) and urine pregnancy tests for premenopausal female participants. In addition, fasting venous blood samples were collected to assess for HbA1c, fasting blood glucose, vitamin E levels, serum biomarkers (TGF-β1, VEGF-A), and safety tests such as renal profile, lipid profile, and liver function test. A nerve conduction study (NCS) was performed on all patients to assess their baseline neurological function. Three nerves were investigated, namely, the upper limbs median sensory, lower limbs sural sensory, and tibial motor nerves. In order to diagnose and assess the patients’ neuropathic pain, they were requested to complete the 12-item self-reported Neuropathic Pain Questionnaire (NPQ) at this visit. Electrocardiograms (ECGs) were conducted to ensure that the patients were fit to participate in the study. In addition, a retinal photograph was also taken using a fundus camera to determine the patients’ retinal pathology at baseline.

#### 2.4.2. Randomization Visit

Upon reviewing the participant’s biochemistry results, they were informed of their eligibility for the study over telephone. Those who met the inclusion and exclusion criteria were scheduled to return to one of the Monash University CRCs for randomization. Investigational products were dispensed according to the patient’s group.

#### 2.4.3. Follow-Up Visits

Study patients were followed up at four weeks to monitor for compliance and adverse events. Compliance to the investigational products was determined by counting the remaining capsules. Anthropometric measurements, blood pressure, capillary fasting blood glucose, and urine FEME were regularly performed at each follow-up visit to monitor the participants. Fasting blood samples were collected for HbA1c assessments and vitamin E measurements during follow-up visits at 2 months and 6 months. NCSs were also conducted at 2 months and 6 months post-intervention. NPQ was assessed again at 6 months post-baseline.

#### 2.4.4. End-of-Study Visit

Study patients came for end-of-study assessments after 12 months of the double-blind treatment period. Adverse events and compliance were monitored as previous visits. Fasting blood and urine samples were taken per screening visit for investigation. NCSs, retinal photographs, and ECGs were also conducted at the end of study. NPQ was assessed again at 12 months post-baseline.

#### 2.4.5. Washout Visit

Study patients were followed up at the CRCs 6 months after the double-blind treatment period. Adverse events were monitored, as with previous visits. Fasting blood and urine samples were taken as per each screening visit for investigation. NCSs, retinal photographs, and ECGs were also conducted as at the end of study.

### 2.5. Outcomes

The primary outcome variables of this study were the mean/median changes in nerve conduction parameters at 2 months, 6 months, and 12 months, and at 6 months post-washout from baseline, as assessed by NCSs.

The secondary outcome variables were the serum biomarkers TGF-β1 and VEGF-A, postulated to be affected in DPN at baseline and the end of study. Adherence of study participants was determined by measuring the fasting venous vitamin E levels.

### 2.6. Nerve Conduction Study

Standardized nerve conduction study (NCS) methodology was followed to conduct NCSs using Medelec Synergy (VIASYS Healthcare, Philadelphia, PA, USA) [14,15]. Three nerves were tested bilaterally on all patients, namely, the median and sural sensory nerves, and the tibial motor nerve. The subject’s skin temperature over the upper and lower limbs was maintained above 32 °C throughout the process. During a sensory NCS, five parameters were reported, which were onset latency, peak latency, sensory nerve action potential (SNAP) amplitudes, conduction velocity (CV), and peak velocity (PV). In contrast, parameters recorded in motor NCSs included distal and proximal onset latencies, compound muscle action potential (CMAP) amplitudes, distance between distal and proximal stimulation points, and CV. The validated methodology by Ng Y.T. et al. was used in this study [13].

To reduce operational bias, a single machine was used at both clinical research centres and the technique to conduct nerve conduction studies (NCSs) was standardized and strictly followed throughout the study under the supervision of a clinical neurologist, Botross N.P. Hence, the NCSs were conducted in the same manner with the same machine by the same standardized method from screening to the washout visit for each patient.

### 2.7. Serum VEGF-A and TGF-β1

Eight millilitres of fasting blood samples were collected from each patient and stored in serum-separating tubes (SST). The samples were then centrifuged with an Eppendorf Centrifuge 5702R (Hamburg, Germany) at 3600 rpm for 15 min on the same day. Following centrifugation, the sera were extracted and aliquoted into separate 1 mL Eppendorf tubes to be flash-frozen and stored at −80 °C. These biomarkers were measured using enzyme-linked immunosorbent assays (ELISAs) with TECAN Infinite 200 PRO (Zurich, Switzerland) and their respective ELISA kits. The ELISA kits for VEGF-A (Elanscience E-EL-H0111, USA) and TGF-β1 (Elabscience E-EL-H5615, USA) had intra-assay coefficient variances of 4% and inter-assay coefficient variances of 8%. To minimize inter-assay variability, measurements of serum biomarkers levels were performed at the end of the study.

### 2.8. Adherence and Vitamin E Levels

Patients were informed to bring their remaining capsules during each follow-up visit. Compliance was determined by counting the remaining capsules that were not consumed. Furthermore, fasting venous vitamin E levels were measured at baseline, and after 2 months, 6 months, and 12 months, to determine the patients’ adherence.

Plasma vitamin E levels were measured for all patients using the high-performance liquid chromatography (HPLC; HPLC 1200, Agilent, Santa Clara, CA, USA) technique with fluorescence detection, developed by Che H. et al. [16], with modifications. The plasma vitamin E levels were corrected for lipids [17].

### 2.9. Renal Profile, Lipid Profile, and Liver Function Tests

Patients’ serum samples were sent to the nationally certified pathology laboratory on the day of collection to test for renal profiles, lipid profiles, and liver function. These tests were performed using Abbott Diagnostics ARCHITECT (Elgin, IL, USA) with coefficient variances of less than 6% in general. Parameters measured for the renal profile were the estimated glomerular filtration rate (eGFR), serum creatinine, and blood urea nitrogen (BUN). Total cholesterol, triglycerides, high-density lipoprotein cholesterol (HDL-C), and low-density lipoprotein cholesterol (LDL-C) were measured for the lipid profile. Liver function tests included assessing aspartate transaminase (AST) and alanine transaminase (ALT).

### 2.10. HbA1c Assessment

Blood samples collected on the day of visit were sent to the nationally certified pathology laboratory for HbA1c measurements. The assessment was performed using a Cobas Integra 400 plus analyzer (Roche Diagnostics, Laval, QC, Canada), with a measuring range of 4.3–18.8% and a coefficient variance of 5%.

### 2.11. Sample Size

The sample size was calculated based on the sural nerve conduction velocity among T2DM patients from a study published by Dunnigan S.K. et al. [18]. A minimum sample size of 36 participants in each arm was sufficient to detect a clinically important difference of 5% difference between groups in improving sural sensory nerve conduction velocity, assuming a standard deviation of 2.2 m/s using an independent t-test to achieve 90% power and a 5% level of significance. Considering a maximum dropout rate of 5%, 38 participants in each arm, giving a total sample size of 76 participants for two treatment arms, was minimally required.

### 2.12. Statistical Analysis

The statistical data analysis was performed using R Studio Version 1.3.1073. The analysis of primary outcomes was conducted on a modified intention-to-treat (mITT) population (all randomized participants who had received at least one dose of investigational product and had at least one post-baseline assessment). The predictive mean matching imputation method was used to handle missing data. Analysis of washout data was performed based on a per-protocol principle, for which only participants who returned after 6 months of washout were included.

Independent t-tests or Mann–Whitney U tests were used to test for differences in continuous variables between the Tocovid and placebo groups, depending on the normality of the variables as assessed by the Kolmogorov–Smirnov test and to determine a 95% CI for the intervention group difference in mean/median response (treatment effect). The between-intervention-group differences in categorical variables were tested with chi-squared test or Fisher’s exact test, depending on their suitability. Paired t-tests or Wilcoxon signed-rank tests were used to test for within-group differences in continuous variables, depending on the normality of the variables. A significance level of 5% was used for all statistical tests, i.e., *p*-values of <0.05 were considered statistically significant.

The left and right sides of each measured nerve were treated independently for analysis. Nerves affected by conditions other than diabetes mellitus and nonresponsive nerves upon stimulation were not included in the analysis. Only patients who were predicted to have neuropathic pain according to the discriminant function score at baseline were included in the analysis of component NPQ scores.

### 2.13. Ethics

The clinical trial was carried out in accordance with the Declaration of Helsinki and the Malaysian Guidelines for Good Clinical Practice (GCP). The study protocol received ethical approval from the Monash University Human Research Ethics Committee (MUHREC) (project number: 12090). This trial was registered in the Australian New Zealand Clinical Trials Registry (ANZCTR), with the trial registration number ACTRN12619001568101.

## 3. Results

A total of 155 T2DM patients were screened over a period of two months: 88 of them who met the inclusion and exclusion criteria were recruited and randomized into either placebo or active groups (200 mg Tocovid BD). The mITT population included 85 participants (n = 43 in the Tocovid group; n = 42 in the placebo group), excluding 3 patients who dropped out before the first follow-up visit. Most (70 of 88) participants completed study treatment for 12 months, because 8 of them were lost at follow-up, 3 withdrew due to adverse events, 1 was unable to return for the end-of-study visit due to personal reasons, and 6 were unable to return for the last follow-up due to movement control orders implemented by the Malaysian government as a measure against the COVID-19 pandemic. Out of the 70 participants who completed 12 months of treatment, 3 were lost to follow-up after 6 months of washout. Five out of the six who failed to return for the final follow-up visit due to COVID-19 returned for the washout visit. Hence, 72 participants were included in the washout analysis. The nerve conduction parameters for median, sural, and tibial nerves were measured bilaterally for all 88 patients. Upon excluding nerves that were affected by conditions apart from diabetes mellitus and unresponsive nerves, sample sizes for median, sural, and tibial nerves were 133, 100, and 149, respectively, for analysis of the 12-month treatment period. On the other hand, the sample sizes in the 6-month washout analysis were 111, 83, and 128 for median, sural, and tibial nerves, respectively. A summary of the patient recruitment flowchart is shown in Figure 1.

### 3.1. Baseline Characteristics

There were no notable differences in baseline characteristics between intervention groups that could potentially affect the efficacy assessment (Table 1). There were more males (65.9%) than females (34.1%) enrolled in this study. The sample also reflected the national demographics of Malaysia: half of the participants were Malay, followed by Chinese (25%), Indian (20.5%), and others (4.5%). The median age of all participants was 64 years old, and on average, the study patients had had diabetes for 15.9 years and HbA1c levels of 7.6%. At baseline, there were no significant differences in the level of serum biomarkers and plasma vitamin E, or baseline safety tests between intervention groups.

The correlation between nerve conduction parameters and general characteristics, as well as biomarkers at baseline, are illustrated in Table 2. The correlation analysis showed that the duration of diabetes was significantly negatively correlated with median and sural nerve conduction parameters. This indicates that participants who developed diabetes for a longer time exhibited a greater reduction in the velocity of sural nerves and amplitudes of both median and sural nerves. HbA1c was found to be negatively correlated with conduction parameters in all nerves, except the distal amplitude at the ankle for tibial nerves. This means that those with higher HbA1c levels at baseline had lower nerve conduction parameters at baseline, i.e., lower amplitude and slower velocity. Age was negatively correlated with tibial nerve action potential amplitude and median nerve action potential, as well as PV. Body mass index (BMI) had a weak but significant negative correlation with the median nerve peak-to-peak amplitude (PPA), whereas systolic blood pressure (SBP) was significantly correlated with tibial nerve action potential (distal amplitude at ankle, AA, and proximal amplitude at knee, KA). Furthermore, there were significant negative correlations between TGFβ-1 and the action potential amplitudes of median and sural sensory nerves, as well as the CV of sural nerves. VEGF-A was negatively associated with sural nerve CV and PV. No correlation was found between DBP and baseline nerve conduction parameters of all three nerves.

### 3.2. Effects Observed during the Intervention Trial

Table 3 shows the mean or median changes in median and sural sensory nerve conduction parameters between Tocovid and placebo groups at 2 months, 6 months, and 12 months post-baseline. The changes in tibial motor nerve conduction parameters between intervention groups are tabulated in Table 4. The changes from baseline to each time point were compared between groups to investigate the efficacy of Tocovid on diabetic peripheral neuropathy. There was evidence of Tocovid efficiency if a significant *p*-value and positive treatment effect were obtained from the between-group comparison (*p* < 0.05). An increase in any of the nerve conduction parameters denoted an improvement in nerve function. Sensory velocity correlates directly with latency [19]; therefore, only sensory velocity was included in our analysis.

At baseline, all the nerve conduction parameters assessed for all three nerves were not statistically different between Tocovid and placebo groups (Table 4).

#### 3.2.1. Two Months of Intervention

There were statistically significant differences in conduction velocity (CV) and peak velocity (PV) between the Tocovid and placebo groups for all three nerves. After two months of Tocovid supplementation, participants in the Tocovid group exhibited improvements of 1.25 m/s (IQR 2.85), 1.4 m/s (IQR 1.60), and 0.6 m/s (IQR 1.88) in CV for median, sural and tibial nerves, respectively. The median CV differences in median, sural, and tibial nerves were 1.80 m/s (95% CI: 1.10, 2.70; *p* < 0.001), 2.20 m/s (95% CI: 1.60, 2.90; *p* < 0.001), and 1.80 m/s (95% CI: 1.10, 2.60; *p* < 0.001) higher than the placebo group.

On the other hand, the intervention group exhibited increments of 1.03 m/s (SD 1.75) and 1.1 m/s (SD 1.57) in peak median and sural nerve velocities from baseline. The mean PV differences for median and sural nerves were 1.38 m/s (95% CI: 0.75, 2.01; *p* < 0.001) and 1.64 m/s (95% CI: 1.00, 2.27; *p* < 0.001) higher in the Tocovid group than with the placebo. Nonetheless, the amplitudes of action potential for all nerves were not statistically different between Tocovid and placebo groups.

#### 3.2.2. Six Months of Intervention

Six months after the beginning of treatment, all velocity parameters continued to show significant improvements from baseline in the Tocovid group, and differences with the placebo remained statistically significant. The treatment effects on CV were 1.60 m/s (95% CI: 1.10, 2.70; *p* < 0.001) for median nerves, 2.10 m/s (95% CI: 1.50, 2.90; *p* < 0.001) for sural nerves, and 1.30 m/s (95% CI: 0.60, 2.20; *p* < 0.001) for tibial nerves. For peak velocity, the treatment effects of median and sural nerves were 1.30 m/s (95% CI: 0.70, 2.00; *p* < 0.001) and 1.41 m/s (95% CI: 0.67, 2.14; *p* < 0.001), respectively. However, the amplitudes of action potentials of all nerves still did not exhibit statistical difference between intervention groups after 6 months of supplementation, except for PP amplitude, which had a treatment effect of 1.70 µV (95% CI: 0.10, 3.20; *p* = 0.040).

#### 3.2.3. Twelve Months of Intervention (End of Treatment)

After 12 months of Tocovid supplementation, all velocity parameters in sural nerves showed statistically significant differences between the Tocovid and placebo groups. The Tocovid group exhibited improvements of 1.80 m/s (IQR 2.05, *p* = 0.036) and 2.5 m/s (IQR 3.6, *p* < 0.001) in the conduction and peak velocity of the sural nerve, respectively. For median nerves, participants in the Tocovid group exhibited significant improvements of 1.97 m/s (IQR 4.28, *p* = 0.007) in conduction velocity, although the peak velocity was not significantly different from the placebo group (*p* = 0.484). The conduction velocities of tibial nerves were not statistically different between intervention groups.

There were no significant differences in HbA1c, blood pressure, serum biomarkers (TGFβ-1 and VEGF-A), or safety tests (eGFR, serum creatinine, and urea) between the Tocovid and placebo groups after 12 months of treatment (Table 5). Glycaemic control between intervention groups was similar throughout the 12-month treatment period, and HbA1c levels in each group were stably maintained below 8% (Appendix A).

#### 3.2.4. Changes in Time within the Intervention Group

The mean/median changes in nerve conduction parameters from baseline within each intervention group at each follow-up visit are illustrated in Appendix A. In the Tocovid group, NP amplitude, PP amplitude, and the peak velocity of the median nerve significantly improved from baseline up to 6 months, but no significant difference was observed after 12 months. Improvements of 1.20 µV (95% CI: 0.05, 2.60; *p* = 0.036), 3.25 µV (95% CI: 0.60, 6.45; *p* = 0.016) and 0.71 m/s (95% CI: 0.19, 1.24; *p* = 0.009) were found in median NP amplitude, PP amplitude, and peak velocity, respectively. After 12 months of Tocovid supplementation, the median nerve conduction velocity of participants in this treatment group increased by 2.08 m/s (95% CI: 1.26, 2.90; *p* < 0.001). On the other hand, there were no significant differences from baseline for all nerve conduction parameters in the placebo group throughout the study period.

For sural nerves, all conduction parameters except NP amplitude were significantly different from baseline in the Tocovid group. Improvements from baseline were seen as early as 2 months after supplementation, and the change persisted until 12 months. At the end of study, differences of 2.10 µV (95% CI: 0.75, 3.40; *p* = 0.004), 1.75 m/s (95% CI: 0.95, 2.39; *p* < 0.001), and 3.00 m/s (95% CI: 2.10, 4.54; *p* < 0.001) were found in the sural PP amplitude, CV, and PV of participants receiving Tocovid. In contrast, the sural conduction velocity of the placebo group decreased by 0.74 m/s (95% CI: −1.23, −0.24; *p* = 0.005) after 6 months. After 2 months, the placebo group exhibited significant decreases in peak velocity from baseline, −0.65 m/s (95% CI: −1.15, −0.05; *p* = 0.041), but significant increases by 0.89 m/s (95% CI: 0.12, 1.66; *p* = 0.024) at the end of study. There were no significant differences from baseline in the sural nerve action potential amplitudes.

Tibial nerve conduction parameters in the Tocovid group demonstrated significant improvements from baseline at all time points, except in conduction velocity comparisons of 12 months with baseline. In the placebo group, conduction velocity significantly decreased by 1.15 m/s (95% CI: −1.86, −0.45; *p* = 0.002) at 2 months post-baseline, and 1.25 m/s (95% CI: −2.20, −0.20; *p* = 0.023) after 12 months. Both Tocovid and placebo groups exhibited significant improvements in the amplitude of action potential.

#### 3.2.5. Adherence and Vitamin E Levels

Compliance throughout the 12-month treatment period, as well as the average compliance over 12 months, was not statistically different between intervention groups. Participants in both groups had good compliance of more than 96% throughout the study (Appendix A).

The adherence of study participants was confirmed by the measurement of plasma vitamin E levels at fasting state, as shown in Appendix A. There were significant differences in the levels of α-, γ-, δ-tocotrienols between intervention groups at 2, 6, and 12 months post-baseline. All three forms of plasma tocotrienol levels were significantly higher among the participants given Tocovid compared to the placebo at the end of 12 months. The levels of α-, γ-, δ-tocotrienols decreased from baseline at 2 months, 6 months, and 12 months in the placebo group. The median differences in α-, γ-, δ-tocotrienol levels were 24.53 pg/mL (95% CI: 6.42, 40.45), 18.13 pg/mL (95% CI: 3.43, 34.43), and 6.62 ng/mL (95% CI: 1.60, 11.36) higher in the Tocovid group than in the placebo group, respectively. The increment in tocotrienols levels can be seen as early as 2 months after supplementation (significant *p*-values were obtained from within-Tocovid-group comparisons with baseline levels). In contrast, both intervention groups exhibited decrements in α-tocopherol levels over the study period, with participants in the placebo group having a greater degree of reduction. The median difference between groups was 0.73 ng/mL (95% CI: 0.24, 1.32) for α-tocopherol levels after 6 months of Tocovid supplementation.

Lipid profiles of study participants were analysed with plasma vitamin E levels corrected for lipids. The results are presented in the Appendix A. There were no significant differences in changes in total cholesterol (TC), triglycerides (TG), high-density lipoprotein cholesterol (HDL-C), or low-density lipoprotein cholesterol (LDL-C) from baseline between intervention groups at 2 months and 6 months. However, levels of HDL-C at 6 months were significantly higher than baseline within each group. The mean increment of HDL-C from baseline was 0.10 mmol/L (95% CI: 0.03, 0.17; *p* = 0.004) in the Tocovid group, whereas HDL-C increased by 0.20 mmol/L (95% CI: 0.01, 0.21; *p* < 0.001) among participants in the placebo group. In contrast, the levels of LDL-C decreased significantly from baseline after 6 months within each group. Study participants who received Tocovid experienced a reduction in LDL-C of 0.30 mmol/L (95% CI: −0.60, −0.05; *p* = 0.017), whereas LDL-C was depleted by 0.30 mmol/L (95% CI: −0.50, −0.10; *p* = 0.009) among participants in the placebo group.

#### 3.2.6. Neuropathic Pain Questionnaire (NPQ)

Only 15 patients (17%) were predicted to have neuropathic pain at baseline, with 8 of them from the Tocovid group (Table 6). These 15 patients were included in the analysis of component NPQ scores. The component scores of 12 items in the NPQ were compared between intervention groups and are tabulated in Table 7. No significant differences were found in component NPQ changes at both 6 and 12 months. However, Tocovid was found to reduce the patients’ numbness (*p* = 0.034) and freezing pain (*p* = 0.026) after 12 months of supplementation.

#### 3.2.7. Adverse Events (AEs)

All randomized patients were included in the report of adverse events (AEs) and serious adverse events (SAEs) (Appendix A). A total of 19 patients reported at least one AE throughout the study period, whereas 5 of them reported an SAE. There were 75 adverse events (AEs) reported throughout the study period, with 41 events (54.7%) occurring in the Tocovid group and 34 events (45.3%) in the placebo group. Gastrointestinal disorders were the most common adverse events reported, accounting for 14.7% of the total AEs reported. Infectious and neurological events were the most frequent events reported in the Tocovid group (six events each), whereas gastrointestinal disorders were the most common AEs in the placebo group, with six events reported.

A total of five serious adverse events (SAEs) were reported in this clinical trial: three in the interventional group and two in the control group. Three patients receiving Tocovid were admitted due to seizure, viral fever, and for renal stone removal. One patient from the placebo group was admitted to hospital for the removal of a tumour in the ureter with partial bladder resection. The other patient from the same group was hospitalized for vaginal hysterectomy to remove a fibroid. All five patients were discharged in good condition, and there was no mortality during the study period.

### 3.3. Subgroup Analysis

#### 3.3.1. Effects of Baseline Conduction Velocity

In order to investigate the effect of baseline structural changes on improvements in nerve conduction, patients were classified into two subgroups according to the mean conduction velocity at baseline due to the absence of a universal standard of normal values for NCS. Between-group comparisons were performed for each nerve (Table 8).

There were no significant differences in median SNAP amplitudes or tibial CMAP amplitudes in both subgroups, which further confirmed our previous findings of no treatment effects in nerve conduction amplitudes. Patients in the interventional group with high-baseline CV exhibited greater effects in nerve conduction velocities for the median sensory nerve as compared to those with low baseline CV, which only exhibited improvements for up to 2 months. Patients in both subgroups presented similar effects of Tocovid in improving sural sensory nerve CV. However, significant improvements in the amplitudes of sural nerve action potential were observed at 12 months in the low-baseline CV subgroup. Reduction in tibial nerve CV was observed in the high-baseline CV subgroup at 6 and 12 months in both intervention groups, with a significantly greater decrease in the placebo group.

#### 3.3.2. Effects of Duration of Diabetes and HbA1c

Patients were separately stratified according to the median duration of diabetes and HbA1c levels for each nerve, and changes in CV at 2, 6, and 12 months were analysed and compared between groups (Table 9). Participants in the Tocovid group who had had diabetes for less than 15 years showed greater median nerve CV improvements compared to the placebo after 12 months of vitamin E supplementation (2.6 m/s, IQR 4.3; *p* = 0.014). Tocovid is equally effective in improving sural and tibial nerve CV for up to 6 months of supplementation, regardless of the duration of diabetes.

Patients in the Tocovid group with poorer glycaemic control exhibited significant improvements in median CV (2.6 m/s, IQR 3.6; *p* = 0.016) at 12 months. In contrast, patients with good glycaemic control experienced improvements in sural CV at the end of the study (1.69 m/s, SD 2.07; *p* = 0.012). Participants with high HbA_1c_ consuming Tocovid exhibited improvements in tibial CV for up to 6 months (1.0 m/s, IQR 3.5; *p* = 0.003).

### 3.4. Six-Month Washout

The mean or median differences in nerve conduction parameters from baseline to after 6 months of washout between the intervention and placebo groups were compared and are tabulated in Table 10. The changes from end of 12 months of vitamin E supplementation to after 6 months of washout were also compared between intervention groups.

There were no significant differences in any nerve conduction parameters from baseline to after 6 months of washout between groups for all three nerves.

For median nerves, the change in nerve conduction velocity (CV) from the end of 12 months of vitamin E supplementation to 6 months of washout was significantly different between groups (*p* = 0.004). There was a slight increase of 0.6 m/s (IQR 1.40) in the placebo groups, whereas a reduction of 0.85 m/s (IQR 2.93) was seen in the Tocovid group after discontinuing the vitamin E supplementation for 6 months. All other nerve conduction parameters did not show significant between-group differences from the end of treatment to 6 months of washout for the median nerve. Significant differences were found by comparing conduction velocities from baseline and the end of the 12-month treatment to 6 months after washout within the placebo group. The median nerve conduction velocity in the placebo group after 6 months of treatment cessation increased by 0.79 m/s (SD 2.74, *p* = 0.033) from baseline, and 0.6 m/s (IQR 1.40, *p* = 0.034) from the end of 12 months of treatment. For the Tocovid group, there was a significant increase from baseline of 1.9 m/s (IQR 3.07, *p* = 0.003), but no significant difference from the end of treatment. This indicates that Tocovid improved the median nerve conduction velocity after 12 months of supplementation and remained stable after 6 months of washout.

For sural nerves, all nerve conduction parameters exhibited significant differences between groups in changes from the end of treatment to 6 months of washout, except the NP amplitude. Participants from the Tocovid group exhibited reductions of 1.45 µV (IQR 3.88, *p* = 0.007), 0.50 m/s (IQR 2.93, *p* < 0.001), and 1.10 m/s (IQR 2.67, *p* < 0.001) in PP amplitude, conduction velocity, and peak velocity, respectively, as compared to the end of the 12-month treatment; these changes were significantly different from the placebo group. Comparisons from the end of treatment to 6 months of washout within the Tocovid group were performed, and statistically significant differences were found in PP amplitude (−1.45 µV, IQR 3.88, *p* = 0.007), conduction velocity (−0.50 m/s, IQR 2.93, *p* < 0.001), and peak velocity (−1.10, IQR 2.67, *p* < 0.001). There were significant increases of 1.45 m/s (IQR 3.38, *p* = 0.023) in sural nerve conduction velocity and 1.02 m/s (SD 2.55, *p* = 0.013) in peak velocity from baseline to 6 months of washout.

There were no significant changes between intervention groups from the end of the 12-month treatment to 6 months of washout found in conduction parameters of the tibial nerve. When we compared within the placebo group, there were significant changes from baseline to 6 months after washout in all three nerve conduction parameters: *p* < 0.001 for conduction velocity, *p* = 0.025 for distal amplitude at ankle, and *p* = 0.001 for proximal amplitude at knee. Within the Tocovid group, comparisons of conduction velocity from baseline and the end of treatment to 6 months of washout were both statistically significant. A negative value of change represented a reduction in conduction velocity. The conduction velocity at washout was statistically lower than the velocity at baseline (−1.34 m/s, SD 3.52, *p* = 0.005) and the end of the 12-month treatment (−1.32 m/s, SD 3.70, *p* = 0.008).

Lipid profiles and plasma vitamin E levels of study participants were also measured and analysed after 6 months of washout (Appendix A). No significant differences were found between Tocovid and placebo groups for TC, TG, HDL-C, and LDL-C. Levels of plasma γ-tocotrienol (*p* = 0.044) and δ-tocotrienol (*p* = 0.047) were significantly higher in the Tocovid group as compared to those receiving the placebo. In both study groups, all plasma vitamin E levels decreased significantly over the 6-month washout period, with participants in the Tocovid group exhibiting greater reductions. Furthermore, participants in both groups exhibited significant decreases in plasma vitamin E levels as compared to the levels at baseline.

### 3.5. Overall Variation in Nerve Conduction Velocities

Figure 2a displays the overall change in median nerve conduction velocity in participants taking Tocovid compared with the placebo group. There was an upward trend in the Tocovid group throughout the 12-month treatment period, and the changes from baseline were significantly higher than in the placebo group. The overall change of sural nerve conduction velocity in participants taking Tocovid compared with the placebo group was illustrated in Figure 2b. For the first 6 months, the Tocovid group showed increments from baseline whereas the conduction velocity in the placebo group decreased. At the end of the study, there were no statistically significant changes in CV from baseline between groups. Tibial nerve conduction velocities in both Tocovid and placebo groups generally showed downward trends from baseline, as shown in Figure 2c. There were no significant differences in nerve CV of all three nerves between Tocovid and placebo groups after 6 months post-washout.

## 4. Discussion

To date, there have been few reports of beneficial effects of vitamin E on diabetic neuropathy. A study by Tutuncu N.B. et al. showed that a supplementation of 900 mg synthetic vitamin E, dl-a-tocopheryl acetate daily for 6 months significantly improved median sensory nerve conduction velocity and tibial motor nerve distal latency [20]. This is the only study to date that had reported objective improvements in nerve function in patients with diabetes. However, this was a preliminary study, with only 21 patients with symptomatic symmetrical distal neuropathy (11 of them receiving vitamin E). Apart from that, several studies have reported subjective assessments which have shown that vitamin E may reduce neuropathic pain symptoms in patients with DPN by utilizing self-reported questionnaires [21,22]. In patients with diabetic neuropathy who received vitamin E-400 capsules for 3 months, 46 out of 92 exhibited reductions in total neuropathy pain scores and improvements in physical scores from the RAND-36 questionnaire as compared to a placebo group [21]. Ogbera A.O. et al. reported that after receiving a combination treatment of vitamin E (400 mg/day, unknown composition) and evening primrose (daily doses from 500–1000 mg) for 2 weeks, 88% of the study patients (70 patients) found themselves to be relieved from “burning pain” [22]. Although these studies [20,21,22] showed encouraging results, it is vital to note that none of these studies explained the true composition of the vitamin E supplementation. In addition, tocopherol has previously been thought to be the most superior isoform of vitamin E, although publications involving tocotrienols only count for around 3% of all publications regarding vitamin E [23].

In this randomized placebo-controlled trial, tocotrienol-rich vitamin E was shown to improve nerve conduction velocities (CV and PV), but with no significant effects on the amplitudes of SNAP or CMAP after 12 months of supplementation. This confirmed the previous preliminary findings which showed the effectiveness of Tocovid in improving velocity parameters of nerves at 8 weeks [13]. There is currently no effective treatment for DPN. Symptomatic relief, glycaemic control, and addressing cardiovascular risk appear to be the mainstays of DPN management [24]. Improvements in nerve velocity parameters shown in this study reflect the effectiveness of vitamin E in repairing the damaged myelin sheath of nerve fibres, which indicates its potential to effectively treat DPN.

However, findings in this study are contrary to the findings of Hor C.P. et al. that reported no significant differences in sensory nerve conduction studies between the treatment group that received 200 mg of mixed tocotrienols twice daily for 12 months and a placebo group [25]. Similarly, the subjective assessments for neuropathic symptoms studied in the study, Total Symptom Score (TSS) and Neuropathy Impairment Score (NIS), were not statistically different between groups at the end of 12 months of study. However, only patients that displayed neuropathic symptoms were recruited for the study, which was different from the current study because we included patients regardless of the presence of neuropathic symptoms. The differences in the patient populations might be the cause of the contradictory findings. Additionally, the HbA1c levels of participants in this study were higher (range: 8.7–9.3%) as compared to the clinical trial presented in this paper (range: 7.34–8%). Hence, the differences in glycaemic control might contribute to the conflicting findings; poor glycaemic control was shown to play an etiological role in DPN [26].

In general, this study illustrated greater improvements in the sensory nerve than motor nerve. Numerous animal models of nerve regeneration have been assayed in the past for studies of nerve repair. However, their findings differ widely from each other. Some studies showed a faster regeneration in sensory fibres, whereas others found motor fibres to regenerate faster, and some showed no differences between sensory and motor nerves in terms of nerve regeneration [27]. However, the variable findings on this topic warrant further research to develop a better understanding of regenerative neurobiology.

It was shown in recent research that whole plantar nerve (WPN) conduction studies are useful for clinicians to recognize very distal nerve damage of the lower limbs in the preclinical phase. This finding is important, because the early diagnosis of DPN can help to reduce the incidence and implement preventive strategies against devastating complications [9].

Diabetic neuropathy was found to be significantly correlated with poor glycaemic control, a longer duration of diabetes, and higher age in various studies [28,29,30]. Our correlation analysis revealed that levels of HbA1c were significantly negatively correlated with all baseline nerve conduction parameters, whereas the duration of diabetes was negatively correlated with SNAP amplitudes of median and sural nerves, as well as sural nerve velocities. In contrast to a study by Booya F. et al. which showed no correlation between diabetic neuropathy and high BP [30], we found that systolic blood pressure was negatively correlated with tibial CMAP amplitudes.

Our previous study also reported no difference in biomarkers of oxidative stress (MDA) and inflammatory activities (VCAM-1 and TNFR-1) after 8 weeks of Tocovid supplementation, despite finding highly significant improvements in nerve conduction velocities. However, the neuronal cell biomarker, serum NGF, was found to be elevated in the Tocovid group after 8 weeks of supplementation, which reflected the reversal of neuronal injuries and restoration of nerve function [13].

In addition, two additional serum biomarkers, TGF-β1 and VEGF-A, were included in this clinical trial to explore possible alternative pathways targeted by Tocovid in improving nerve conduction velocities. TGF-β1 was found to induce cellular injury in diabetic neuropathy in an experimental setting, which suggested its possible implication in the pathogenesis of DPN. High levels of serum TGF-β1 were detected in diabetic patients with peripheral neuropathy, and the levels increased with the duration of diabetes [31]. A study by Ybarra J. et al. reported a highly significant positive correlation between DPN and TGF-β1, and logTGF-β1 was higher in neuropathy patients compared to normal [32]. Findings from these studies have suggested TGF-β1 as a potential biomarker for DPN. However, there were no significant differences between the levels of serum TGF-β1 of treatment and placebo groups in this clinical trial after 12 months of supplementation, despite significant improvements in nerve velocities.

Vascular factors were implicated in the pathogenesis of diabetic neuropathy. As opposed to the harmful effects of VEGF on diabetic retinopathy and nephropathy, VEGF-A has been shown to possess neuroprotective effects [33,34,35,36], promote neuronal growth [37,38,39], and relieve diabetic neuropathic symptoms [40]. However, serum VEGF levels were found to be higher in patients with DPN than those without DPN, and the levels were particularly high at the stage where patients displayed neuropathic symptoms [41]. In this clinical trial, there were no significant differences in the levels of serum VEGF between intervention groups at baseline as well as the end of study. This suggests that significant improvements in nerve conduction after the supplementation of Tocovid were not through the upregulation of VEGF.

None of the biomarkers measured in our previous study and this clinical trial showed significant changes, despite the significant improvement seen in nerve conduction velocities. This indicates that other possible pathways are regulated by tocotrienol-rich vitamin E. In this case, proteomics appears to be an ideal tool to fill the knowledge gap about other molecular mechanisms that can be regulated by tocotrienol-rich vitamin E. Hence, further studies are warranted to assess the complexity of vitamin E and explore the exact effect mechanisms of tocotrienol-rich vitamin E on improving DPN.

Levels of plasma vitamin E were significantly higher in participants receiving Tocovid as compared to the placebo. This confirms the compliance of the participants in this clinical trial. However, plasma vitamin E levels in the placebo group decreased significantly over the study period. In addition, the safety tests performed in this study illustrated that there were no parameters with significant statistical differences between both intervention groups, implying a low side effect profile with the FDA-approved recommended dosage of tocotrienols. Nevertheless, significant reductions in LDL-C and increments in HDL-C from baseline were evident within each intervention group at 6 months post-intervention. This may suggest the presence of the Hawthorne effect, which is defined as a change in behaviour resulting from the awareness of being observed. Having awareness of their participation in the study, all study participants were more health-conscious during the study period, consuming diets lower in dietary fats. A longitudinal clinical study that investigated the impact of the Hawthorne effect in anaesthesia practice found that participants who were informed of their inclusion in the study reported better postoperative quality of life compared to unaware patients [42]. It is generally believed that dietary fat is necessary for the absorption of vitamin E due to its lipophilic properties. Studies have demonstrated a better absorption of vitamin E with higher levels of fat in the diet [43,44,45]. Hence, the steady decrease in plasma α-tocopherol levels within each intervention group might have been due to reduced dietary fat intake during the clinical trial period. Although the levels of α-tocopherol were reduced in the Tocovid group, the reduction was to a lesser extent as compared to those receiving the placebo. This further confirmed the patented and bioenhanced property of Tocovid Suprabio^TM^—an increased absorption of tocotrienols and tocopherols. In addition, this could be attributable to the amount of α-tocopherol in the Tocovid soft gel. On the other hand, levels of α-, γ-, and δ-tocotrienol remained high in the Tocovid group, despite decreasing LDL-C levels owing to the tocotrienol-rich vitamin E supplementation and the probable distinct metabolic pathway of tocotrienol as compared to tocopherol. An animal study which studied tissue levels of tocotrienols in tocopherol-transport protein (TTP)-deficient mice conducted by Khanna S. et al. supports this hypothesis [46]. It was revealed that orally supplemented tocotrienols were found in important organs of TTP-deficient mice after the long-term supplementation of tocotrienols, with the skin, adipose, ovaries, and heart being the preferred destinations [46]. Hence, this suggests that the transportation of tocotrienols is through a TTP-independent mechanism [46]. Overall, the relationship between dietary fat intake and the absorption of vitamin E should be well-understood to increase the bioavailability of supplementary vitamin E.

This study, however, is subject to several limitations. First, the levels of neurotoxic amino acid, homocysteine, were not measured in this clinical trial. Hyperhomocysteinemia was found to be an independent risk factor of peripheral neuropathy in several studies [47,48,49]. Hence, it would be necessary to include homocysteine levels in future studies of DPN to explore its potential interactions with the effect of pharmacological interventions on DPN. Second, the patient population enrolled in this study suffered from type 2 diabetes mellitus, and not necessarily from diabetic neuropathy. Further research is needed to study the effects of tocotrienol-rich vitamin E on the targeted population with diabetic peripheral neuropathy. Lastly, F-wave latency was not measured in this study. The F-wave latency has been shown to be the most reproducible and sensitive measure for the study of nerve pathology in comparison to other parameters, namely, the latency, action potential amplitude, and conduction velocity [50,51]. In particular, tibial F-wave minimal latency appeared to be one of the best techniques to detect early DPN. In addition, the inclusion of F-wave minimal latency in conventional NCSs can detect up to 98% of electrophysiologic abnormalities [52]. Therefore, it should be included in future electrodiagnostic tests for DPN.

## 5. Conclusions

This is the final report on the double-blind, placebo-controlled, multicentre, randomized controlled clinical trial, which showed that improvements in nerve conduction velocity in median and sural sensory nerves persist for up to 12 months, whereas improvements in the tibial motor nerve were only observed until 6 months post-baseline. The effects on nerve conduction velocity diminished after 6 months of washout.

## Figures and Tables

**Figure 1 nutrients-13-03770-f001:**
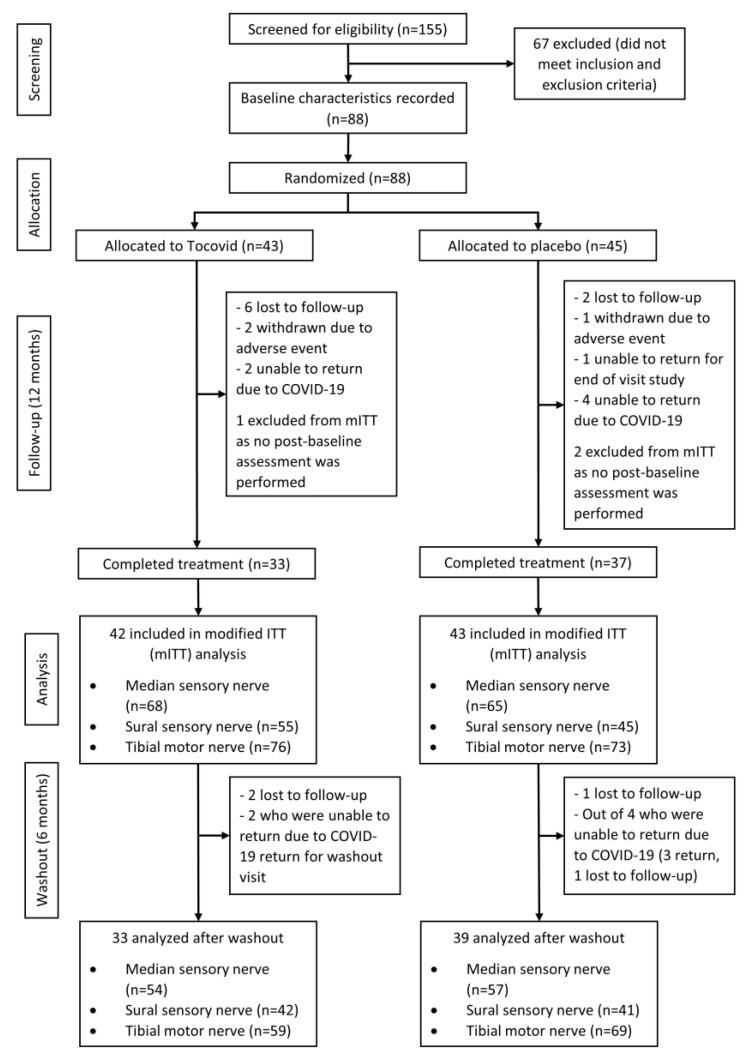
Summary of patient recruitment and study flow diagram.

**Figure 2 nutrients-13-03770-f002:**
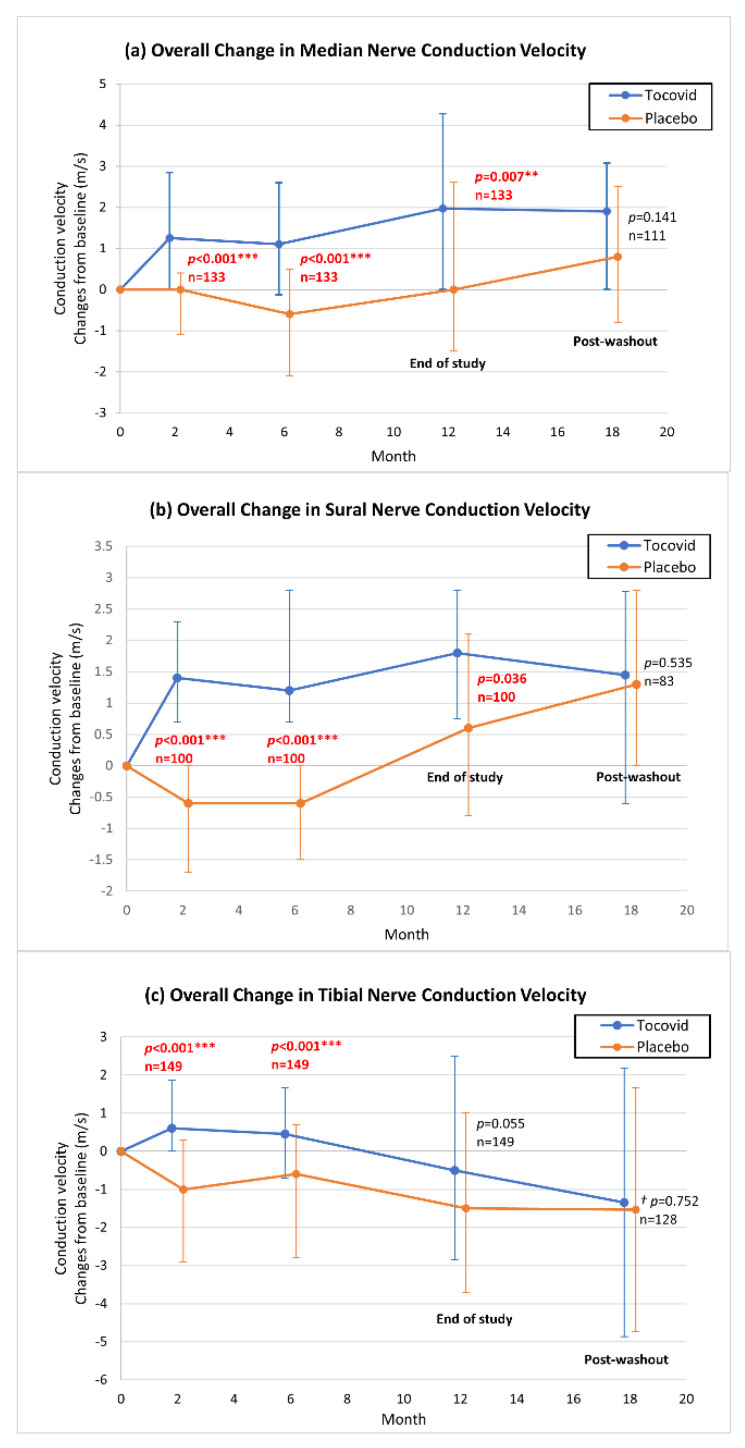
Graph of overall change in (**a**) median, (**b**) sural, and (**c**) tibial nerve conduction velocities. ** *p* < 0.01, *** *p* < 0.001. All data are presented as the median, with error bar representing the inter-quartile range. ^†^ Data are presented as the mean with error bars representing standard deviations.

**Table 1 nutrients-13-03770-t001:** Baseline characteristics of study patients.

Baseline Characteristics	Total (n = 88)	Tocovid (n = 43)	Placebo (n = 45)	*p*-Value
Gender, n (%)				
Male	58 (65.9)	29 (67.4)	29 (64.4)	0.943 ^†^
Female	30 (34.1)	14 (32.6)	16 (35.6)	
Race, n (%)				
Malay	44 (50)	23 (53.5)	21 (46.7)	0.834 ^‡^
Chinese	22 (25)	10 (23.3)	12 (26.7)	
Indian	18 (20.5)	9 (20.9)	9 (20)	
Others	4 (4.5)	1 (2.3)	3 (6.7)	
Age (years) ^¶^	64 (13)	63 (11.5)	64 (13)	0.997
Duration of DM (years) ^§^	15.9 ± 8.8	15.5 ± 8.57	16.2 ± 9.06	0.720
HbA1c (%) ^§^	7.6 ± 1.1	7.56 ± 1.00	7.7 ± 1.2	0.546
SBP (mmHg) ^§^	131.4 ± 12.9	134.7 ± 12.9	129.2 ± 12.6	0.101
DBP (mmHg) ^§^	77.8 ± 8.69	78.0 ± 8.45	77.5 ± 9.01	0.775
BMI (kg/m^2^) ^§^	28.2 ± 4.64	28.0 ± 4.13	28.3 ± 5.11	0.811
Serum Biomarkers				
TGFβ-1 (ng/mL)	33.1 (31.1)	33.6 (16.7)	32.1 (37.4)	0.887
VEGF-A (pg/mL)	707 (600)	762 (478)	655 (582)	0.231
Plasma Vitamin E				
α-Tocotrienol (pg/mL) ^¶^	37.3 (55.5)	36.7 (53.8)	37.5 (56.4)	0.729
γ-Tocotrienol (pg/mL) ^¶^	24.3 (29.4)	19.6 (30.8)	42.7 (29.5)	0.815
δ-Tocotrienol (pg/mL) ^¶^	12.5 (13.6)	12.4 (16.2)	12.7 (13.2)	0.784
α-Tocopherol (ng/mL) ^¶^	1.69 (1.46)	1.63 (1.20)	2.02 (1.58)	0.698
Safety Tests				
eGFR (mL/min/1.73 m^2^) ^¶^	66.3 (36.9)	64.5 (35.2)	69 (37.5)	0.848
Serum Creatinine (µmol/L) ^¶^	99.5 (43.2)	95.1 (46.6)	100.5 (38.5)	0.710
Urea (mmol/L) ^¶^	5.9 (3.3)	5.7 (3.6)	6.1 (3.2)	0.844
AST (U/L) ^¶^	10 (11)	10 (11.5)	11 (9)	0.391
ALT (U/L) ^¶^	14 (14.3)	12 (15.5)	14 (12)	0.358
TC (mmol/L) ^§^	34.1 ± 18.4	37 ± 19.1	31.3 ± 17.5	0.146
HDL-C (mmol/L) ^§^	12.8 ± 4.9	12.9 ± 5.0	12.8 ± 4.9	0.920

^†^ Chi-squared test. ^‡^ Fisher’s exact test. More than 20% of the cells had expected values of less than 5. ^§^ Data are presented as the mean ± standard deviation; *p*-value obtained from independent t-tests by comparing the mean between Tocovid and placebo groups. ^¶^ Data are presented as the median (interquartile range); *p*-value obtained from Mann–Whitney U test by comparing the median between Tocovid and placebo groups. Abbreviations: DM, diabetes mellitus; HbA1c, haemoglobin A1c; SBP, systolic blood pressure; DBP, diastolic blood pressure; BMI, body mass index; TGFβ-1, transforming growth factor beta-1; VEGF-A, vascular endothelial growth factor A; eGFR, estimated glomerular filtration rate; AST, aspartate transaminase; ALT, alanine transaminase; TC, total cholesterol; HDL-C, high-density lipoprotein cholesterol.

**Table 2 nutrients-13-03770-t002:** Correlation between baseline characteristics and biomarkers with nerve conduction parameters at baseline.

Baseline Characteristics	Pearson’s Correlation (r)										
	Median				Sural				Tibial		
	NPA	PPA	CV	PV	NPA	PPA	CV	PV	CV	AA	KA
Age (years)	−0.25 **	−0.22 **	−0.15	−0.17 *	−0.11	−0.14	−0.06	−0.11	0.06	−0.25 **	−0.21 *
DM duration (years)	−0.27 **	−0.28 **	−0.16	−0.14	−0.51 ***	−0.42 ***	−0.39 ***	−0.32 **	−0.05	−0.15	−0.12
BMI (kg/m^2^)	−0.17	−0.17 *	−0.10	−0.07	−0.01	−0.17	0	−0.01	0.08	−0.10	−0.13
HbA1c (%)	−0.26 **	−0.29 ***	−0.27 **	−0.28 **	−0.23 *	−0.22 *	−0.33 ***	−0.33 ***	−0.47 ***	−0.14	−0.24 **
SBP (mmHg)	−0.12	−0.11	−0.08	−0.10	−0.03	−0.06	−0.01	−0.04	−0.14	−0.28 ***	−0.20 *
DBP (mmHg)	0.03	−0.01	0.02	0.02	−0.08	−0.11	−0.05	−0.05	−0.10	−0.09	−0.05
TGFβ-1 (ng/mL)	−0.17 *	−0.18 *	−0.03	−0.01	−0.16 *	−0.18 *	−0.21 *	−0.14	0.02	−0.11	−0.11
VEGF-A (pg/mL)	0.04	0.05	0.01	0.04	−0.06	−0.06	−0.34 ***	−0.29 **	−0.09	0.10	0.06

* *p* < 0.05, ** *p* < 0.01, *** *p* < 0.001. Abbreviations: NPA, negative-to-peak amplitude (µV); PPA, peak-to-peak amplitude (µV); CV, conduction velocity (m/s); PV, peak velocity (m/s); AA, distal amplitude at ankle (µV); KA, proximal amplitude at knee (µV); DM, diabetes mellitus; BMI, body mass index; HbA1c, haemoglobin A1c; SBP, systolic blood pressure; DBP, diastolic blood pressure; TGFβ-1, transforming growth factor beta-1; VEGF-A, vascular endothelial growth factor A.

**Table 3 nutrients-13-03770-t003:** Comparison of median and sural sensory nerve conduction parameters between Tocovid and placebo groups.

	Median Sensory Nerve				Sural Sensory Nerve			
	Tocovid (n = 68) ^†^	Placebo (n = 65) ^†^	Treatment Effect (95% CI)	*p*-Value ^‡^	Tocovid (n = 55) ^†^	Placebo (n = 45) ^†^	Treatment Effect (95% CI)	*p*-Value ^‡^
NP Amplitude (µV)								
At baseline	19.5 (14.0)	17.5 (12.0)	2.3 (−1.1, 5.7)	0.200	9.7 (8.00)	10.1 (4.9)	0.20 (−1.80, 2.30)	0.843
2-month changes	1.25 ± 5.98	0.79 ± 4.87	0.45 (−1.42, 2.33)	0.632 ^§^	0.2 (4.45)	−0.4 (1.9)	0.60 (−0.70, 1.80)	0.362
6-month changes	0.8 (6.35)	0.5 (6.20)	0.5 (−1.1, 2.1)	0.539	0.7 (4.35)	−1.2 (3.70)	1.40 (−0.10, 2.70)	0.075
12-month changes	0.25 (5.72)	0.6 (5.9)	−0.2 (−2.0, 1.7)	0.843	−0.6 (5.15)	−0.3 (4.50)	0.50 (−1.60, 2.10)	0.635
PP Amplitude (µV)								
At baseline	31.9 (22.7)	27.3 (14.7)	3.5 (−2.0, 8.8)	0.213	6.6 (8.3)	8.7 (7.8)	−1.40 (−3.50, 0.90)	0.244
2-month changes	3.15 (10.6)	0.9 (6.9)	2.0 (−0.8, 5.0)	0.137	1.11 ± 2.48	0.43 ± 2.99	0.68 (−0.41, 1.76)	0.218 ^§^
6-month changes	3.45 (14.1)	0.6 (7.20)	2.1 (−1.1, 5.7)	0.193	1.70 (5.13)	0.40 (5.30)	1.70 (0.10, 3.20)	0.040 *
12-month changes	3.60 (15.4)	2.60 (13.8)	2.0 (−1.6, 6.2)	0.320	2.10 (5.35)	1.70 (4.56)	1.00 (−0.74, 2.80)	0.257
Conduction Velocity (m/s)								
At baseline	44.0 ± 6.79	43.2 ± 7.10	0.88 (−1.51, 3.26)	0.469 ^§^	43.8 (7.8)	43.1 (6.7)	0 (−2.40, 2.60)	0.906
2-month changes	1.25 (2.85)	0 (1.5)	1.80 (1.10, 2.70)	<0.001 ***	1.4 (1.60)	−0.6 (1.7)	2.20 (1.60, 2.90)	<0.001 ***
6-month changes	1.10 (2.73)	−0.6 (2.6)	1.60 (0.70, 2.40)	<0.001 ***	1.2 (2.1)	−0.6 (1.5)	2.10 (1.50, 2.90)	<0.001 ***
12-month changes	1.97 (4.28)	0 (4.1)	1.60 (0.50, 2.60)	0.007 **	1.80 (2.05)	0.6 (2.90)	1.97 (1.10, 3.45)	0.036 *
Peak Velocity (m/s)								
At baseline	35.2 ± 4.80	34.7 ± 5.06	0.47 (−1.22, 2.16)	0.582 ^§^	34.1 ± 4.32	34.4 ± 3.73	−0.23 (−1.85, 1.40)	0.783 ^§^
2-month changes	1.03 ± 1.75	−0.35 ± 1.92	1.38 (0.75, 2.01)	<0.001 ***^,§^	1.1 ± 1.57	−0.54 ± 1.62	1.64 (1.00, 2.27)	<0.001 ***^,§^
6-month changes	0.70 (2.62)	−0.5 (2.10)	1.30 (0.70, 2.00)	<0.001 **	1.36 ± 1.97	−0.04 ± 1.68	1.41 (0.67, 2.14)	<0.001 ***^,§^
12-month changes	0.75 (3.90)	0 (2.10)	0.30 (−0.60, 1.10)	0.484	2.5 (3.6)	0.20 (3.1)	2.10 (1.00, 3.20)	<0.001 ***

* *p* < 0.05, ** *p* < 0.01, *** *p* < 0.001. ^†^ Data are presented as the mean ± standard deviation or median (interquartile range); ^‡^
*p*-value obtained from Mann–Whitney U tests by comparing the median between Tocovid and placebo groups. ^§^
*p*-value obtained from independent t-tests by comparing the mean between Tocovid and placebo groups. Abbreviations: M, mean; MD, median; 95% CI, 95% confidence interval.

**Table 4 nutrients-13-03770-t004:** Comparison of tibial motor nerve conduction parameters between Tocovid and placebo groups.

	Tibial Motor Nerve			
	Tocovid (n = 76) ^†^	Placebo (n = 73) ^†^	Treatment Effect (95% CI)	*p*-Value ^‡^
Conduction Velocity (m/s)				
At baseline	41.5 ± 5.54	39.9 ± 6.15	1.60 (−0.30, 3.49)	0.097 ^§^
2-month changes	0.6 (1.88)	−1 (3.2)	1.80 (1.10, 2.60)	<0.001 ***
6-month changes	0.45 (2.38)	−0.6 (3.50)	1.30 (0.60, 2.20)	<0.001 ***
12-month changes	−0.5 (5.35)	−1.5 (4.70)	1.20 (−9.71 × 10^−6^, 2.46)	0.055
Distal Amplitude at Ankle (mV)				
At baseline	7.25 (5.75)	7.6 (7.2)	0.6 (−0.90, 2.10)	0.387
2-month changes	0.8 (1.85)	0.6 (1.80)	0.20 (−0.20, 0.70)	0.295
6-month changes	0.35 (2.62)	0.6 (1.7)	−0.30 (−0.90, 0.30)	0.359
12-month changes	1.60 (3.25)	0.5 (1.70)	0.60 (−0.20, 1.34)	0.122
Proximal Amplitude at Knee (mV)				
At baseline	5.20 (3.98)	5.5 (5.90)	0.40 (−0.70, 1.60)	0.475
2-month changes	0.60 (1.25)	0.5 (1.40)	0.10 (−0.30, 0.40)	0.653
6-month changes	0.5 (1.52)	0.5 (1.5)	−0.10 (−0.50, 0.30)	0.779
12-month changes	0.3 (2)	0.2 (1.6)	0.19 (−0.34, 0.70)	0.555

*** *p* < 0.001. ^†^ Data are presented as the mean ± standard deviation or median (interquartile range); ^‡^
*p*-value obtained from Mann–Whitney U tests by comparing the median between Tocovid and placebo groups. ^§^
*p*-value obtained from independent t-tests by comparing the mean between Tocovid and placebo groups. Abbreviations: M, mean; MD, median; 95% CI, 95% confidence interval.

**Table 5 nutrients-13-03770-t005:** Comparison of analytes between intervention and placebo groups at the end of study.

Analytes	Tocovid (n = 42)	Placebo (n = 43)	Mean/Median Difference (95% CI)	*p*-Value
HbA1c (%) ^‡^	7.34 (1.20)	8 (1.91)	−0.50 (−1.00, 4.47 × 10^−5^)	0.058
SBP (mmHg) ^†^	135.97 ± 11.91	132.70 ± 13.14	3.27 (−2.15, 8.68)	0.233
DBP (mmHg) ^†^	74.39 ± 8.10	74.93 ± 7.73	−0.54 (−3.96, 2.87)	0.752
Serum Biomarkers				
TGFβ-1 (ng/mL) ^‡^	22.0 (11.5)	17.8 (13.3)	2.81 (−0.99, 6.36)	0.115
VEGF-A (pg/mL) ^‡^	745.1 (505.3)	726.1 (715.9)	10.4 (−177.2, 207.9)	0.916
Safety Tests				
eGFR ^†^	65.9 ± 18.7	64.0 ± 21.2	1.89 (−6.74, 10.52)	0.664
Serum Creatinine ^‡^	95.9 (38.7)	100.8 (33.7)	−2.85 (−15.0, 9.73)	0.613
Urea ^‡^	6.18 (2.98)	6.35 (3.04)	−0.50 (−1.34, 0.39)	0.287

^†^ Data are presented as the mean ± standard deviation; *p*-value obtained from independent t-tests by comparing the mean between Tocovid and placebo groups. ^‡^ Data are presented as the median (interquartile range); *p*-value obtained from Mann–Whitney U tests by comparing the median between Tocovid and placebo groups. Abbreviations: HbA1c, haemoglobin A1c; SBP, systolic blood pressure; DBP, diastolic blood pressure; TGFβ-1, transforming growth factor beta-1; VEGF-A, vascular endothelial growth factor A; eGFR, estimated glomerular filtration rate.

**Table 6 nutrients-13-03770-t006:** Neuropathic Pain Questionnaire (NPQ) scores at baseline.

	Tocovid (n = 43)	Placebo (n = 45)	*p*-Value ^†^
Neuropathic pain	8	7	0.923
Non-neuropathic pain	35	38	

Patients were predicted to have neuropathic or non-neuropathic pain based on the scoring worksheet of the NPQ, which calculated the discriminant function score. Discriminant function scores below 0 predict non-neuropathic pain, whereas discriminant function scores at or above 0 predict neuropathic pain. ^†^ Chi-squared test.

**Table 7 nutrients-13-03770-t007:** Component scores of the Neuropathic Pain Questionnaire (NPQ) by intervention group.

Component	Tocovid (n = 8)		Placebo (n = 7)		
	Mean ± SD/ Median (IQR)	*p*-Value (within Group) ^†^	Mean ± SD/ Median (IQR)	*p*-Value (within Group) ^†^	*p*-Value (between Group) ^‡^
Burning pain					
At baseline	20.5 (65.0)		40 (55)		0.810
At 6 months	18.8 (35.0)	0.361	40 (45)	0.789	
At 12 months	0 (0)	0.058	0 (0)	0.100	
6-month changes	−7.93 ± 21.74		−3.57 ± 34.0		0.769 ^¶^
12-month changes	−19.5 ± 65.0		−30 ± 45		0.765 ^¶^
Overly sensitive to touch					
At baseline	12.5 (48.5)		0 (75)		0.660
At 6 months	7 (21.25)	0.174	30 (60)	1.00	
At 12 months	0 (0)	0.100	0 (25)	0.462	
6-month changes	0 (30)		0 (0)		0.334
12-month changes	−12.5 (41.0)		0 (50)		0.951
Shooting pain					
At baseline	40 ± 27.3/40 (35)		0 (40)		0.229
At 6 months	26.4 ± 24.7	0.199 ^§^	0 (0)	1.00	
At 12 months	14 (50)	0.107	0 (30)	0.850	
6-month changes	0 (41.5)		0 (0)		0.948
12-month changes	−14.0 ± 21.8		−2.86 ± 61.6		0.663 ^¶^
Numbness					
At baseline	53.5 ± 22.1		52.9 ± 30.4/50 (20)		0.963 ^¶^
At 6 months	39.5 ± 29.4	0.124 ^§^	48.6 ± 25.4	0.200 ^§^	
At 12 months	29.8 ± 29.3	0.034 *^,§^	60.0 (12.5)	0.892	
6-month changes	−4 (15)		0 (5)		0.433
12-month changes	−23.8 ± 25.6		−2.14 ± 17.3		0.082 ^¶^
Electric pain					
At baseline	9.0 (22.5)		0 (0)		0.168
At 6 months	3.0 (22.5)	0.371	0 (0)	NA	
At 12 months	0.0 (24.5)	1.00	0 (0)	1.00	
6-month changes	0 (3)		0 (0)		0.204
12-month changes	0 (19)		5.7 (25.1)		0.660
Tingling pain					
At baseline	32.1 ± 14.1/30 (5.25)		41.4 ± 15.7/50 (20)		0.249 ^¶^
At 6 months	28.0 ± 17.8	0.298 ^§^	44.3 ± 12.7	0.569 ^§^	
At 12 months	0 (23.5)	0.313	0 (40)	0.289	
6-month changes	0 (8.5)		0 (0)		0.433
12-month changes	−14.1 ± 34.7		−20.0 ± 35.6		0.749 ^¶^
Squeezing pain					
At baseline	0 (6.25)		0 (25)		0.710
At 6 months	0 (10)	0.586	0 (15)	1.00	
At 12 months	0 (0)	0.371	0 (0)	0.346	
6-month changes	0.0 (2.5)		0 (0)		1.00
12-month changes	0.00 (6.25)		0 (25)		0.71
Freezing pain					
At baseline	24.6 ± 24.8/20 (21)		0 (30)		
At 6 months	17.4 ± 19.6	0.353 ^§^	0 (30)	NA	
At 12 months	0 ± 0	0.026 *^,§^	0 (0)	0.181	
6-month changes	0.00 (3.75)		0 (0)		0.620
12-month changes	−20 (21)		0 (30)		0.546
How unpleasant is your usual pain?					
At baseline	38.8 ± 20.8		57.1 ± 30.9		0.195
At 6 months	28.1 ± 19.6	0.164 ^§^	52.9 ± 28.1	0.200 ^§^	
At 12 months	31.5 ± 29.6	0.396 ^§^	37.9 ± 28.0	0.144 ^§^	
6-month changes	0.0 (12.5)		0 (5)		0.679
12-month changes	−9 (20)		−5 (25)		0.954
How overwhelming is your usual pain?					
At baseline	43.1 ± 28.1		30.0 ± 31.1/20 (45)		0.406
At 6 months	36.1 ± 26.2	0.491 ^§^	27.1 ± 26.3	0.356 ^§^	
At 12 months	29.3 ± 29.1	0.076 ^§^	0 (55)	0.490	
6-month changes	0.0 (2.5)		0 (0)		1.00
12-month changes	−13.9 ± 18.8		−4.29 ± 12.7		0.276 ^¶^
Increased pain due to touch					
At baseline	2.5 (15.0)		0 (0)		0.045 *
At 6 months	0 (10)	0.423	0 (0)	NA	
At 12 months	0 (0)	0.201	0 (0)	1.00	
6-month changes	0.0 (2.5)		0 (0)		0.620
12-month changes	0 (15)		0 (0)		0.191
Increased pain due to weather changes					
At baseline	29.4 ± 26.2/25 (45)		0 (0)		0.044 *
At 6 months	14.5 ± 10.5	0.114 ^§^	0 (0)	1.00	
At 12 months	22.1 ± 25.6	0.577 ^§^	0 (0)	1.00	
6-month changes	0.0 (32.5)		0 (0)		0.535
12-month changes	0 (45)		0 (0)		0.951

* *p* < 0.05. Data are presented as the median (interquartile range) or mean ± standard deviation. ^†^
*p*-value obtained from Wilcoxon signed-rank tests by comparing the median within each intervention group. ^‡^
*p*-value obtained from Mann–Whitney U tests by comparing the median between Tocovid and placebo groups. ^§^
*p*-value obtained from paired t-tests by comparing the mean within each intervention group. ^¶^
*p*-value obtained from independent t-tests by comparing the mean between Tocovid and placebo groups. Abbreviations: NA, not applicable.

**Table 8 nutrients-13-03770-t008:** Subgroup analysis comparing changes in nerve conduction parameters between intervention groups, stratified according to baseline conduction velocity (CV).

Baseline Characteristics		Tocovid (n)	Placebo (n)	2-Month Changes			6-Month Changes			12-Month Changes		
				Tocovid	Placebo	*p*-Value ^†^	Tocovid	Placebo	*p*-Value ^†^	Tocovid	Placebo	*p*-Value ^†^
Low- baseline CV	Median CV < 43.61 m/s ^a^											
	CV	36	32	1.60 (2.25)	0.20 (2.88)	0.008 **	1.75 (2.13)	0.20 (3.25)	0.096	2.95 (4.42)	2.55 (3.03)	0.370
	PV			1.07 ± 1.47	0.32 ± 1.84	0.067 ^‡^	0.95 (2.20)	0 (2.93)	0.037 *	0.67 ± 2.24	1.37 ± 2.23	0.203 ^‡^
	Sural CV < 43.16 m/s ^a^											
	NPA	26	25	0.45 ± 2.02	−0.62 ± 2.66	0.111 ^‡^	0.36 ± 2.90	−1.58 ± 3.13	0.026 *^,^^‡^	1.82 ± 3.55	−0.99 ± 3.83	0.009 **^,^^‡^
	PPA			0.79 ± 2.12	0.19 ± 2.65	0.380 ^‡^	1.70 (4.85)	−0.5 (4.5)	0.070	3.74 (5.12)	1.0 (5.5)	0.014 *
	CV			1.85 (1.28)	0.0 (1.3)	<0.001 ***	1.7 (2.3)	0 (0.7)	<0.001 ***	2.95 ± 3.23	1.59 ± 2.55	0.102 ^‡^
	PV			1.35 (1.48)	0.0 (1.5)	<0.001 ***	2.29 ± 1.79	0.43 ± 1.73	<0.001 ***^,^^‡^	3.20 (6.84)	0.4 (3.4)	<0.001 ***
	Tibial CV < 40.71 m/s ^a^											
	CV	38	40	1.05 (2.50)	−0.3 (1.9)	<0.001 ***	0.85 (3.20)	0 (1.83)	0.015 *	1.78 ± 4.27	0.39 ± 4.21	0.152 ^‡^
	AA			0.80 (1.88)	0.40 (1.25)	0.276	0.00 (2.75)	0.55 (1.48)	0.291	1.95 (3.18)	0.25 (1.37)	0.016 *
High- baseline CV	Median CV ≥ 43.61 m/s ^a^											
	CV	32	33	0.95 (3.00)	−0.7 (3.10)	<0.001 ***	0.35 (2.95)	−1.2 (2.3)	0.008 **	0.9 (3.30)	−0.8 (2.3)	0.005 **
	PV			0.80 (2.95)	−0.5 (1.4)	<0.001 ***	0.26 ± 2.30	−1.22 ± 1.67	0.004 **^,^^‡^	0.55 (4.20)	−0.5 (2.3)	0.203
	Sural CV ≥ 43.16 m/s ^a^											
	NPA	29	20	0.1 (5.7)	−0.25 (1.95)	0.684	0.3 (4.8)	0.00 (5.65)	0.927	−3.12 ± 6.09	0.18 ± 4.83	0.049 *^,^^‡^
	CV			1.23 ± 1.70	−1.80 ± 1.47	<0.001 ***^,^^‡^	0.91 ± 2.13	−1.50 ± 1.53	<0.001 ***^,^^‡^	1.0 (3.3)	−0.65 (2.63)	0.065
	PV			0.79 ± 1.59	−1.36 ± 1.56	<0.001 ***^,^^‡^	0.53 ± 1.76	−0.64 ± 1.44	0.019 *^,^^‡^	2.12 ± 2.70	0.30 ± 2.64	0.023 *^,^^‡^
	Tibial CV ≥ 40.71 m/s ^a^											
	CV	38	33	0.40 (1.45)	−2.5 (3.0)	<0.001 ***	−0.35 ± 2.31	−2.21 ± 2.62	0.002 **^,^^‡^	−2.00 (3.18)	−3.30 (4.18)	0.032 *

* *p* < 0.05, ** *p* < 0.01, *** *p* < 0.001. Data are presented as the median (interquartile range) or mean ± standard deviation. Only parameters with significant *p*-values are presented. Complete results of subgroup analysis stratified by baseline CV can be obtained in the Appendix A. ^a^ Mean CV; ^†^
*p*-value obtained from Mann–Whitney U tests by comparing the median between Tocovid and placebo groups. ^‡^
*p*-value obtained from independent t-tests by comparing the mean between Tocovid and placebo groups. Abbreviations: n, number of participants in each intervention group; CV, conduction velocity; PV, peak velocity; NPA, negative-to-peak amplitude; PPA, peak-to-peak amplitude.

**Table 9 nutrients-13-03770-t009:** Subgroup analysis comparing changes in conduction velocity (CV) stratified according to the duration of diabetes and HbA1c.

Baseline Characteristics	Total (N)	Tocovid (n)	Placebo (n)	2-Month Changes			6-Month Changes			12-Month Changes		
				Tocovid	Placebo	*p*-Value ^†^	Tocovid	Placebo	*p*-Value ^†^	Tocovid	Placebo	*p*-Value ^†^
DM duration ^a^												
Median												
<15 years	66	37	29	1.87 ± 2.11	0.06 ± 2.08	<0.001 ***^‡^	1.1 (3.7)	−0.6 (0.9)	0.058	2.6 (4.3)	0.0 (3.6)	0.014 *
≥15 years	67	21	36	1.52 ± 2.05	−0.79 ± 2.94	<0.001 ***^‡^	1.27 ± 2.22	−0.58 ± 2.85	0.005 **^‡^	1.2 (3.5)	0.3 (4.8)	0.129
Sural												
<14 years	47	23	24	1.20 (1.25)	−0.75 (2.10)	<0.001 ***	1.18 ± 2.21	−1.05 ± 1.60	<0.001 ***^‡^	1.0 (3.5)	0 (3.18)	0.394
≥14 years	53	32	21	1.55 ± 1.65	−0.17 ± 1.89	<0.001 ***^‡^	1.45 (2.40)	−0.6 (1.2)	<0.001 ***	2.25 (1.83)	0.7 (3.5)	0.107
Tibial												
<14 years	67	33	34	1.0 (2.4)	−0.3 (3.38)	<0.001 ***	1.1 (2.8)	−0.60 (2.95)	0.004 **	0.76 (5.10)	−0.86 (4.23)	0.153
≥14 years	82	43	39	0.40 (1.65)	−1.3 (3.0)	<0.001 ***	0.3 (2.0)	−0.70 (3.75)	0.032 *	−1.2 (4.75)	−2.4 (5.4)	0.118
HbA1c ^a^												
Median												
<7.55%	65	39	26	0.90 (2.25)	0 (2.03)	0.009 **	0.90 (2.55)	−0.30 (2.05)	0.030 *	2.15 ± 3.20	0.75 ± 4.45	0.145 ^‡^
≥7.55%	68	29	39	1.9 (2.5)	0 (1.8)	<0.001 ***	1.3 (3.3)	−0.60 (2.95)	<0.001 ***	2.6 (3.6)	0 (3.6)	0.016 *
Sural												
<7.2%	48	27	21	1.63 ± 1.86	−0.95 ± 1.21	<0.001 ***^‡^	1.87 ± 2.34	−0.86 ± 1.33	<0.001 ***^‡^	1.69 ± 2.07	−0.13 ± 2.76	0.012 *^,^^‡^
≥7.2%	52	28	24	1.30 (1.55)	−0.6 (2.6)	<0.001 ***	1.30 (1.56)	−0.60 (1.23)	<0.001 ***	1.91 (3.72)	1.05 (3.83)	0.497
Tibial												
<7.55%	74	43	31	0.4 (2.0)	−0.80 (3.25)	0.007 **	0.3 (2.1)	−0.9 (4.7)	0.057	−0.34 ± 2.92	−1.19 ± 3.71	0.274
≥7.55%	75	33	42	0.8 (2.4)	−1.15 (3.18)	<0.001 ***	1.0 (3.5)	−0.30 (2.88)	0.003 **	0.20 (6.98)	−1.60 (4.58)	0.147

* *p* < 0.05, ** *p* < 0.01, *** *p* < 0.001. Data are presented as the median (interquartile range) or mean ± standard deviation. ^a^ Stratified according to the median duration of diabetes; ^†^
*p*-value obtained from Mann–Whitney U tests by comparing the median between Tocovid and placebo groups; ^‡^
*p*-value obtained from independent t-tests by comparing the mean between Tocovid and placebo groups. Abbreviations: N, number of participants in each subgroup for each nerve; n, number of participants in each intervention group; DM, diabetes mellitus; HbA1c, haemoglobin A1c.

**Table 10 nutrients-13-03770-t010:** Comparison of nerve conduction parameters between intervention groups at 6 months post-washout.

	M ± SD/ MD (IQR)	Difference M/MD (95% CI) ^†^	*p*-Value (within Group) ^†^	M ± SD/ MD (IQR)	Difference M/MD (95% CI) ^†^	*p*-Value (within Groups) ^†^	*p*-Value (between Groups) ^‡^
**Median Sensory Nerve**	**Tocovid (n = 54)**			**Placebo (n = 57)**			
NPA (µV)							
At baseline (B)	18.2 (14.4)			17.2 (10.5)			0.368 ^¶^
WO—B	0.45 (5.48)	0.85 (−0.40, 1.90) ^§^	0.188 ^§^	0.2 (4.20)	0.10 (−0.90, 1.00) ^§^	0.830 ^§^	0.362 ^¶^
WO—12M	−0.25 (4.68)	0.16 (−0.95, 1.45) ^§^	0.750 ^§^	−0.4 (6)	−0.50 (−2.05, 0.95) ^§^	0.451 ^§^	0.433 ^¶^
PPA (µV)							
At baseline (B)	31.8 (20.8)			27.3 (14.8)			0.320 ^¶^
WO—B	2.1 (11.2)	3.21 (0.48, 5.93)	0.022 *	1 (6.9)	0.65 (−1.20, 2.25) ^§^	0.533 ^§^	0.256 ^¶^
WO—12M	0.20 (8.38)	−0.49 (−2.64, 1.66)	0.650	−0.1 (8.44)	−0.20 (−2.40, 2.15) ^§^	0.874 ^§^	0.894 ^¶^
CV (m/s)							
At baseline (B)	43.6 ± 6.96			43.4 ± 7.22			0.919
WO—B	1.9 (3.07)	1.75 (0.70, 2.55) ^§^	0.003**^,^^§^	0.80 (3.3)	0.79 (0.07, 1.52)	0.033 *	0.141 ^¶^
WO—12M	−0.85 (2.93)	−0.80 (−1.45, 4.78 × 10^−5^) ^§^	0.051 ^§^	0.6 (1.40)	0.80 (0.05, 1.30) ^§^	0.034 *^,^^§^	0.004 **^,¶^
PV (m/s)							
At baseline (B)	34.9 ± 4.97			34.9 ± 5.05			0.942
WO—B	1.15 (3.47)	0.90 (1.88 × 10^−5^, 1.65) ^§^	0.050 ^§^	0.5 (2.3)	0.35 (−0.14, 0.85)	0.161	0.198 ^¶^
WO—12M	−0.35 (2.35)	−0.25 (−0.75, 0.25) ^§^	0.308 ^§^	0 (1.80)	0.25 (−0.20, 0.70) ^§^	0.259 ^§^	0.128 ^¶^
**Sural Sensory Nerve**	**Tocovid (n = 42)**			**Placebo (n = 41)**			
NPA (µV)							
At baseline (B)	9.35 (7.55)			9.8 (5.4)			0.827 ^¶^
WO—B	−1.62 ± 4.22	−1.55 (−2.75, −0.15) ^§^	0.028 *^,^^§^	−0.66 ± 3.10	−1.00 (−2.00, 0.15) ^§^	0.076 ^§^	0.256
WO—12M	−0.8 (4.37)	−1.10 (−2.60, 0.20) ^§^	0.088 ^§^	−0.3 (3.8)	−0.25 (−1.25, 0.90) ^§^	0.669 ^§^	0.310 ^¶^
PPA (µV)							
At baseline (B)	6.15 (7.3)			7.4 (7.9)			0.310 ^¶^
WO—B	0.788 ± 3.41	0.75 (−0.45, 1.85) ^§^	0.232 ^§^	0.77 ± 4.21	0.70 (−0.65, 2.15) ^§^	0.279 ^§^	0.986
WO—12M	−1.45 (3.88)	−1.65 (−2.82, −0.50) ^§^	0.007 **^,^^§^	0.2 (3.9)	0.15 (−0.95, 1.50) ^§^	0.777 ^§^	0.027 *^,¶^
CV (m/s)							
At baseline (B)	43.3 (8.42)			41.8 (6.7)			0.626 ^¶^
WO—B	1.45 (3.38)	1.15 (0.20, 2.10) ^§^	0.023 *^,^^§^	1.30 (2.8)	1.26 (0.27, 2.25)	0.014 *	0.535 ^¶^
WO—12M	−0.50 (2.93)	−1.70 (−2.50, −0.75) ^§^	<0.001 ***^,^^§^	0.90 (2.20)	0.81 (−0.27, 1.89)	0.136	<0.001 ***^,¶^
PV (m/s)							
At baseline (B)	33.8 ± 4.39			34.2 ± 3.84			0.630
WO—B	0.95 (3.12)	1.02 (0.23, 1.82)	0.013 *	1.1 (1.6)	1.42 (0.58, 2.25)	0.001 **	0.802 ^¶^
WO—12M	−1.10 (2.67)	−1.40 (−2.25, −0.65) ^§^	<0.001 ***^,^^§^	0.70 (2.2)	0.78 (0.05, 1.52)	0.038 *	<0.001 ***^,¶^
**Tibial Motor Nerve**	**Tocovid (n = 59)**			**Placebo (n = 69)**			
CV (m/s)							
At baseline (B)	41.7 ± 5.63			39.8 ± 6.32			0.087
WO—B	−1.34 ± 3.52	−1.34 (−2.26, −0.43)	0.005 **	−1.53 ± 3.20	−1.53 (−2.30, −0.76)	<0.001 ***	0.752
WO—12M	−0.5 (2.65)	−1.32 (−2.29, −0.36)	0.008 **	−0.3 (2.9)	−0.40 (−1.10, 0.30) ^§^	0.275 ^§^	0.368 ^¶^
AA (mV)							
At baseline (B)	6.9 (5.15)			7.2 (7.30)			0.340 ^¶^
WO—B	0.10 (2.25)	0.25 (−0.30, 0.80) ^§^	0.384 ^§^	0.30 (2.10)	0.50 (0.10, 0.95) ^§^	0.025 *^,^^§^	0.384 ^¶^
WO—12M	−0.4 (2.1)	−0.55 (−1.05, −0.05) ^§^	0.028 *^,^^§^	−0.1 (1.6)	−0.10 (−0.50, 0.25) ^§^	0.504 ^§^	0.181 ^¶^
KA (mV)							
At baseline (B)	5.3 (3.9)			5.2 (5.6)			0.236 ^¶^
WO—B	0.10 (1.30)	0.10 (−0.25, 0.50) ^§^	0.512 ^§^	0.30 (1.1)	0.45 (0.15, 0.75) ^§^	0.001 **^,^^§^	0.125 ^¶^
WO—12M	−0.2 (1.60)	−0.30 (−0.65, 0.02) ^§^	0.071 ^§^	0 (1.20)	−0.05 (−0.35, 0.20) ^§^	0.700 ^§^	0.248 ^¶^

* *p* < 0.05, ** *p* < 0.01, *** *p* < 0.001. ^†^ Reported as mean difference (95% CI), *p*-value obtained from paired t-tests by comparing the mean within each intervention group. ^‡^
*p*-value obtained from independent t-tests by comparing the mean between Tocovid and placebo groups. ^§^ Reported as the median difference (95% CI), *p*-value obtained from Wilcoxon signed-rank test by comparing the median within each intervention group. ^¶^
*p*-value obtained from Mann–Whitney U tests by comparing the median between Tocovid and placebo groups. Abbreviations: M: mean; SD, standard deviation; MD, median; IQR, interquartile-range; NPA, negative-to-peak amplitude; PPA, peak-to-peak amplitude; CV, conduction velocity; PV, peak velocity; AA, distal amplitude at ankle; KA, proximal amplitude at knee; WO, washout; 12M, at 12 months.

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
