# Peer review of "Tocotrienol-Rich Vitamin E (Tocovid) Improved Nerve Conduction Velocity in Type 2 Diabetes Mellitus Patients in a Phase II Double-Blind, Randomized Controlled Clinical Trial"

_nutrients, 2021, doi:10.3390/nu13113770_

Round 1

Reviewer 1 Report

In this multicentre, prospective, double-blinded, randomized, placebo-controlled, phase II clinical trial, Chuar et al. investigated the effects of tocotrienol-rich vitamin E vs placebo during 12 months of treatment on nerve conduction parameters and serum biomarkers among patients with type 2 diabetes.

Authors report that after 12 months of supplementation, patients in the tocotrienol-rich vitamin E group showed highly significant improvement in conduction velocity of both median and sural sensory nerves as compared to those in the placebo group. A significant difference in peak velocity was also observed in sural nerve.

This is an interesting paper performed with a very strong methodology (to note that authors evaluated the effect of tocotrienol-rich vitamin E also after 6 months of washout).

I have only minor comments:

Line 123: please report when supplementation was taken (at fasting? Every 12 hours? After meal?).

Line 200: 8ml.....eight ml.

Line 325: subtitle: effect observed during the intervention trial.

Discussion: should be shortened

Lines 629-640: this sentence should be reported in the description of study design. At beginning of discussion please report the main findings. The same in the conclusion (please remove lines 807-810)

Reviewer 2 Report

I enjoyed reviewing this manuscript.

The study is interesting and methodologically correct. The conclusions are supported by results.

The tables and figures are clear. This trial is not entirely original, but it builds on the results of a previous study that are enriched in this study.

I raise a few issues that authors need to address.

Materials and Methods

1- In line 116 it is written ‘unstable eye diseases’. It is not clear what the authors mean. Maybe they mean proliferative retinopathy? Why should this be a contraindication?  Please clarify.

Discussion

2- In line 50, authors state ‘The early diagnosis of diabetic peripheral neuropathy has been difficult as half of the 50 patients do not display any symptoms’. Actually, an early diagnosis of DPN is essential to implement prevention strategies against possible several disabling complications as ulcers and amputations events. DPN, due to its pathophysiological mechanisms, is known to be typically symmetrical and distal. Therefore, it mainly affects the long nerves of the lower limbs. The ability to recognize very distal nerve conduction damage in the preclinical phase is a challenge for the clinician. Very recently, it was observed that whole plantar nerve (WPN) conduction study is an extremely useful tool for an early detection of distal diabetic peripheral neuropathy (Diabetes Res Clin Pract. 2021 Jun;176:108856. doi: 10.1016/j.diabres.2021.108856.). This intriguing issue should be pointed out in the text (introduction as well as discussion) by authors.

3- Figure 2 (Graph of overall variation in median, sural and tibial nerve conduction velocity) and comments on it should be included in the results rather than in the discussion.

Conclusions

4. In the conclusions the authors summarize the results of their study and its originality with respect to the previous pilot study by Ng YT et al. This was already done at the beginning of the discussion. In the discussion, instead of repeating previously exposed concepts, the authors should comment on what the clinical perspectives of this trial may be.

References

5- Many references are around 20 years old. If possible, the majority should be replaced with more recent references.

6- The manuscript needs a linguistic revision of a native English speaker.

Author Response

This manuscript is a resubmission of an earlier submission. The following is a list of the peer review reports and author responses from that submission.

Round 1

Reviewer 1 Report

In the research article entitled, "Tocotrienol-Rich Vitamin E (Tocovid) Improves Diabetic Neuropathy in A Phase II Double-Blind, Randomized Controlled Clinical Trial", by Chuar et al the authors present patient data from a phase II double-blind, randomized controlled clinical trial. The authors claim the improvement of diabetic neuropathy based on nerve conductance test results.

Major comments:

  1. Hyperhomocysteinemia exerts neurotoxic effects and correlates with the development of neuropathy. Homocysteine level is independently associated with the development and severity of DPN. Therefore, it is very surprising to see that homocysteine levels are missing from the study. It would be ideal to report the baseline homocysteine levels and at subsequent follow-up of the patients. 
  2. The assessment of neuropathic symptoms in the inclusion criteria should be of prime importance. In line 4 the authors mention "Up to 53% of patients with DPN experience neuropathic pain-reducing their quality of life and work 45 productivity [4]." But strangely the assessment of neuropathic pain is nowhere to be mentioned in the study. Neither at the baseline nor at subsequent follow-ups. The patients should have been assessed during the screening visit with either Total Symptom Score (TSS) (Further characterized into Lancinating pain, Burning pain, Paresthesia, and asleep numbness) and/or Neuropathy Impairment Score (NIS).
  3. The severe adverse events have not been reported, it would be ideal if the authors list the common adverse events reported by the study group.
  4. Since the entire argument of improvement in diabetic neuropathy in the study is based on improved nerve conduction test results, the methodology of the nerve conduction test is paramount. Therefore, in this study, the role of the operator and interpreter is very crucial for an unbiased nerve conduction test result. The authors should clearly state if the NCTs were conducted in an unbiased way by a single operator, preferably at a single center and interpreted by a clinical neurologist. All other precautions taken into account to make sure these results do not include any operational or interpretational bias should be highlighted.

Author Response

Dear reviewer, please find the point-by-point response from the authors in the attachment. Thank you.

Reviewer 2 Report

This was an interesting continuation study of an 8-week trial to determine if Tocovid improves DPN.  Overall, I believe the research has merit but I have some suggestions/comments to improve the paper.

19 missing a before the word nerve

31 & 32 instead of in 6 months or in 12 months, use at   Also, you say 400mg/d when I thought you were treating with 200 mg/d

42 it should be non not none

44 DPN would have a cause, PN may not

63 inter?  That does not describe the mechanism properly.  Moreover, I would suggest citing the mechanism paper rather than the review.

66 either cytokines or a cytokine

97 T2DM not defined throughout, put in keywords or in text

104 smoker should be smokers

296, 337, 408, 421, 437, 533, 538, 542 Error message.  I'm concerned with what is missing in that text and can not approve this paper until I see the text that should be replacing the error.

356 times not time

391 profiles not profile

592 missing a period at the end of the sentence

616 "important organs"  what does that mean?  Clarify.  Also the interpretation could be the tocotrienols are I the organs due to lack of TTP from how it was written.  Justify

621 Seems like an afterthought.  Why is it more sensitive, what advantage does it have?  It appears the title of the 2 referenced papers were used but no substance.

Table 4 What does tibial data use difference instead of treatment effect like the median and sural nerve data ?

S6 & S7 indicated the vitamin E levels were adjusted for lipids but the lipids were only measured to 6 months.  How can this be?  It's not clear from the table legend in S6 that vitamin E was normalized.  This needs to be clarified.

The levels of vitamin E dropping over time in the placebo patients are very curious. Vitamin E is found in healthy fats. Are there any references to the fact individuals eat better during a study?

It would be nice to see the vitamins E levels after the 6 months wash out.

Author Response

(The authors gave the same response as above.)

Round 2

Reviewer 1 Report

Comments:

  1. I would still point out that the heading of the article can be misleading. Refereeing to the inclusion criteria mentioned in the study, the inclusion of patients is based on the patient population having type 2 diabetes mellitus and not necessarily with symptoms of diabetic neuropathy. Given the fact that no Symptom Score (TSS) and/or neuropathy impairment score was done during inclusion, which raises serious questions on the fact that all patient’s population enrolled were necessarily not suffering from diabetic neuropathy but type 2 diabetes mellitus. As a matter of fact, not all type 2 diabetes mellitus patients would be suffering from diabetic neuropathy.

In light of the above-mentioned points, authors should rephrase the title of the study to, "Tocotrienol-Rich Vitamin E (Tocovid) Improves Nerve Conduction Velocity in Type 2 Diabetes Mellitus Patients in A Phase II Double-Blind, Randomized Controlled Clinical Trial".

  1. In section “2.6. Nerve Conduction Study mention”, include in the manuscript these words now stated by the authors, “to reduce operational bias, a single machine was used at both clinical research centres and the technique to conduct nerve conduction study (NCS) was standardized and strictly followed throughout the study under the supervision of a clinical neurologist, Botross NP. Hence, the NCS was conducted in the same manner with the same machine by the same standardized method from screening to the washout visit on each patient”.

  1. The authors now report data from the patients’ neuropathic pain at the screening. This is an important set of data and should be included in the main manuscript and not in the supplemental. The authors now report, “However, Tocovid was found to reduce the patients’ numbness (p = 0.034) and freezing pain (p = 0.026) after 12 months of supplementation.” However, the quantitative data used for the analysis is not reported. This is an important set of data and a table with all the data sets should be reported in the manuscript. The authors should include a table with all the 12 items in the self-reported Neuropathic Pain Questionnaire (NPQ) and its scoring or the quantitative data.

  1. The authors should highlight all the current limitations of the study, to name a few are:
    1. Homocysteine level is independently associated with the development and severity of DPN, not included in the clinical trial.
    2. The patient population enrolled was suffering from type 2 diabetes mellitus and not necessarily from diabetic neuropathy.

Author Response

Dear reviewer, please see attachment. Thank you.
